# Cyclic di-GMP as an antitoxin regulates bacterial genome stability and antibiotic persistence in biofilms

Hebin Liao[1,2,3†], Xiaodan Yan[1,2†], Chenyi Wang[1,2], Chun Huang[1,2], Wei Zhang[1,2], Leyi Xiao[1,2], Jun Jiang[4], Yongjia Bao[4], Tao Huang[4], Hanbo Zhang[4], Chunming Guo[4], Yufeng Zhang[1,2,5], Yingying Pu[1,2,6]*

[1]The State Key Laboratory Breeding Base of Basic Science of Stomatology & Key Laboratory of Oral Biomedicine Ministry of Education, School & Hospital of Stomatology, Medical Research Institute, Wuhan University, Wuhan, China; [2]Frontier Science Center for Immunology and Metabolism, Wuhan University, Wuhan, China; [3]Translational Medicine Research Center, North Sichuan Medical College, Nanchong, China; [4]Center for Life Sciences, School of Life Sciences, Yunnan University, Kunming, China; [5]Taikang Center for Life and Medical Sciences, Wuhan University, Wuhan, China; [6]Department of Immunology, Hubei Province Key Laboratory of Allergy and Immunology, State Key Laboratory of Virology and Medical Research Institute, Wuhan University School of Basic Medical Sciences, Wuhan, China

*For correspondence:
yingyingpu@whu.edu.cn

†These authors contributed equally to this work

Competing interest: The authors declare that no competing interests exist.

**Abstract** Biofilms are complex bacterial communities characterized by a high persister prevalence, which contributes to chronic and relapsing infections. Historically, persister formation in biofilms has been linked to constraints imposed by their dense structures. However, we observed an elevated persister frequency accompanying the stage of cell adhesion, marking the onset of biofilm development. Subsequent mechanistic studies uncovered a comparable type of toxin-antitoxin (TA) module (TA-like system) triggered by cell adhesion, which is responsible for this elevation. In this module, the toxin HipH acts as a genotoxic deoxyribonuclease, inducing DNA double strand breaks and genome instability. While the second messenger c-di-GMP functions as the antitoxin, exerting control over HipH expression and activity. The dynamic interplay between c-di-GMP and HipH levels emerges as a crucial determinant governing genome stability and persister generation within biofilms. These findings unveil a unique TA system, where small molecules act as the antitoxin, outlining a biofilm-specific molecular mechanism influencing genome stability and antibiotic persistence, with potential implications for treating biofilm infections.

## eLife assessment

This work describes how the toxin-antitoxin (TA) system, which uses the cyclic di-GMP as an antitoxin, controls both the persistence of antibiotics linked to biofilms and the integrity of the bacterial genome. The authors present **solid** evidence linking cyclic di-GMP and the toxin HipH. The work is **valuable** because it establishes the relationship between bacterial persistence and biofilm resilience, which lays a strong basis for future research on the formation of bacterial biofilms and antibiotic resistance.

## Introduction

The prevalence of chronic and recurrent microbial infections has been significantly attributed to the formation of biofilms - complex, surface-attached bacterial communities (*Costerton et al., 1999*; *Hall-Stoodley et al., 2004*; *Lewis, 2005*). In biofilms, bacteria exhibit reduced antibiotic sensitivities, a phenomenon occurring independently of the presence of resistance determinants on genetic elements (*Ceri et al., 1999*). This antibiotic insensitivity is linked to the elevated prevalence of persister cells in biofilms, with rates ranging from 10 to 1000 times higher compared to their planktonic counterparts (*Lewis, 2005*; *Yan and Bassler, 2019*). Persister cells, recognized as phenotypic variants in an isogenic population, demonstrate a remarkable ability to withstand antibiotic treatment and, intriguingly, can resuscitate once the antibiotic pressure is alleviated (*Balaban et al., 2004*; *Bigger, 1944*; *Lewis, 2007*; *Zhou et al., 2023*). These characteristics collectively position persister cells as key contributors to the perpetuation of persistent infections of biofilms.

Despite the high frequency in biofilms and critical roles in both clinical context and evolution processes, the mechanisms underlying persister formation in biofilms are the subject of intense debate. The effects induced by high density, including nutrient and oxygen deficiency, and quorum sensing within dense structure of developed biofilms, have long been considered as primary factors leading to persister formation (*Ganesan et al., 2016*; *Nguyen et al., 2011*; *Yan and Bassler, 2019*; *Yan et al., 2018*). This association originally stems from the hypothesis of limitations in nutrient, oxygen, or antibiotic penetration into the dense structures. Nevertheless, recent challenges to this hypothesis have arisen from studies indicating that the matrix mesh size of biofilms is significantly larger than the size of molecules such as nutrients, oxygen, and antibiotics (*Ganesan et al., 2016*; *Yan and Bassler, 2019*; *Yan et al., 2018*). Notably, most antibiotics do not strongly interact with biofilm matrix components (*Spoering and Lewis, 2001*). An alternative perspective emerges from observations that cells in mature biofilms exhibit metabolic activity similar to cells in the stationary phase (*Nguyen et al., 2011*; *Walters et al., 2003*; *Waters et al., 2016*), during which the rate of persister cell formation significantly surpasses that of cells in the exponential phase. This phenomenon is likely attributed to nutrient exhaustion in the local environment surrounding mature biofilms (*Nguyen et al., 2011*; *Walters et al., 2003*; *Waters et al., 2016*). Meanwhile, some findings propose the involvement of specific genes in persister cell formation in biofilms (*Zhao et al., 2013*), many of these observations are derived from research conducted under planktonic (free-swimming) conditions with bulk cells (*Yan and Bassler, 2019*).

Recent research has shed light on the critical role of small metabolic molecules in regulating the formation and resuscitation of persister cells, including persister cells in biofilms, providing valuable insights into the complex mechanisms of bacterial persistence. Guanosine pentaphosphate and tetraphosphate ((p)ppGpp), key alarmones in the bacterial stringent response, have been shown to mediate ribosome dimerization, leading to persistence, while their degradation triggers resuscitation. This dual function of (p)ppGpp underscores its significance in both initiating and terminating the persister state (*Song and Wood, 2020*; *Wood and Song, 2020*; *Wood et al., 2019*). Additionally, the interaction between cyclic AMP (cAMP) and the cAMP receptor protein (CRP) has been identified as a crucial regulator of bacterial persistence. The cAMP-CRP complex acts as a global regulator, influencing persistence through modulation of cellular metabolism and regulation of stress responses (*Mok et al., 2015*; *Niu et al., 2024*). These molecular mechanisms offer a more nuanced understanding of persister cell dynamics.

Studying a biofilm during its early developmental stage offers a unique vantage point, minimizing the impact of dense structure or nutrient exhaustion, thus allowing for a controlled exploration of intrinsic factors influencing persister cell formation. Cell-surface adhesion is the critical initiation step in the early stage of biofilm development, encompassing both reversible and irreversible processes (*Berne et al., 2018*; *Tolker-Nielsen, 2015*). During reversible adhesion, planktonic bacteria utilize surface appendages such as flagella and pili, fostering a weak attachment through van der Waals forces (*Ren et al., 2018*). Subsequently, these reversible adhered cells sense environmental cues linked to the surface, activating signaling pathways that lead to the synthesis of 3',5'-cyclic diguanylic acid (c-di-GMP). Elevated c-di-GMP levels function as a secondary messenger, orchestrating a shift in bacterial behavior by influencing the expression of genes crucial to biofilm development. This signaling cascade propels the transition from reversible to irreversible adhesion, wherein bacteria

establish microcolonies on the surface, solidifying a more stable and irreversible connection (*Jenal et al., 2017*; *Valentini and Filloux, 2016*).

To explore the intrinsic factors influencing persister cell formation during the early stages of biofilm development, we employed a static biofilm system for our investigation (*Figure 1A*). This system is widely employed in research to scrutinize the pivotal processes of initial adherence during biofilm formation (*Biswas and Mettlach, 2019*; *Merritt et al., 2005*; *Seviour et al., 2021*; *Sung et al., 2022*). In our initial study, we noted an increased persister frequency coinciding with the step of cell adhesion. This observation prompted the hypothesis that elevated c-di-GMP levels in individual cells potentially facilitate the formation of persister cells within biofilms. Further investigation substantiated this hypothesis, demonstrating that the elevation in c-di-GMP indeed promotes the formation of persister cells. Our detailed analysis uncovered the role of c-di-GMP as an antitoxin, exerting control over both the expression and genotoxicity of HipH, thereby regulating genome stability and persister cell formation. These findings propose a biofilm-specific mechanism for antibiotic persistence. Notably, targeting this toxin-antitoxin system effectively inhibited drug tolerance in Uropathogenic *Escherichia coli* (UPEC) infections, presenting promising therapeutic strategies against chronic and relapsing infections.

## Results

### An increased propensity of persister formation upon cell adhesion

To enhance the clinical relevance of our study, we employed UPEC, the predominant cause of urinary tract infections (UTIs) and responsible for approximately 80% of cases. UPEC has a notable ability to form biofilms on various surfaces, including catheter materials, bladder epithelial cell walls, and within bladder cells (Supplementary Information). In our investigation, we selected the clinical isolate UPEC16. This strain demonstrates a significant phylogenetic similarity to established UPEC reference strains such as CFT073 and UTI89 (*Figure 1—figure supplement 1A*). Moreover, it displays sensitivity to a wide range of antibiotics and possesses the capability for surface adhesion. To culture UPEC16 cells in the static biofilm system, exponential phase cells with an optical density (OD600) of 0.3 were diluted in 10-fold in fresh media, followed by static incubation in microtiter dishes for varying time intervals. Planktonic bacteria were then gently washed away, leaving adhered cells for further investigation. In the persister counting assay, both the remaining adhered cells and the removed planktonic cells were exposed to bactericidal antibiotics (ampicillin, 150 μg/ml, 10-fold of MIC) for varying durations. Subsequently, persister counting was conducted through plating.

Biphasic killing curves revealed distinct patterns. When cells were subjected to the same duration of static culture, the persister rate within the adhered population notably surpassed that observed in the planktonic population (*Figure 1B–D*). Additionally, the persister rate within adhered populations exhibited a positive correlation with the prolonged duration of static culture (*Figure 1B–D*). In contrast, the persister rate within planktonic populations displayed little variation with an extended duration of static culture (*Figure 1B–D*). These observations underscore the crucial role of cell adhesion in promoting persister cell formation during the early stages of biofilm development. This raises questions about established ideas linking persister formation mainly with the dense structure of biofilms, suggesting a need to reconsider the factors that influence persister dynamics during biofilm initiation.

### The correlation between heightened c-di-GMP levels and increased persister frequency

The increase in c-di-GMP levels is a distinctive feature of cell-surface adhesion, marking the transition of cells from a planktonic lifestyle to a sessile one (*Fernandez et al., 2020*; *Jones et al., 2015*; *Valentini and Filloux, 2016*). Building on our observations of increased persister rates upon cell adhesion, we assumed that elevated c-di-GMP levels might play a role in promoting persister cell formation. To investigate this hypothesis and discern the relationship between c-di-GMP levels and persister rates, we introduced a ratiometric fluorescent c-di-GMP sensing system (*Vrabioiu and Berg, 2022*) into the UPEC16 strain. MrkH is a transcription factor derived from *Klebsiella pneumoniae*. The system utilizes the conformational changes that occur in MrkH upon binding to c-di-GMP by strategically inserting it into mVenus[NB] between residues Y145 and N146, resulting in the creation of the biosensor referred to as mVenus[NB]-MrkH. This strategic integration results in a ratiometrically

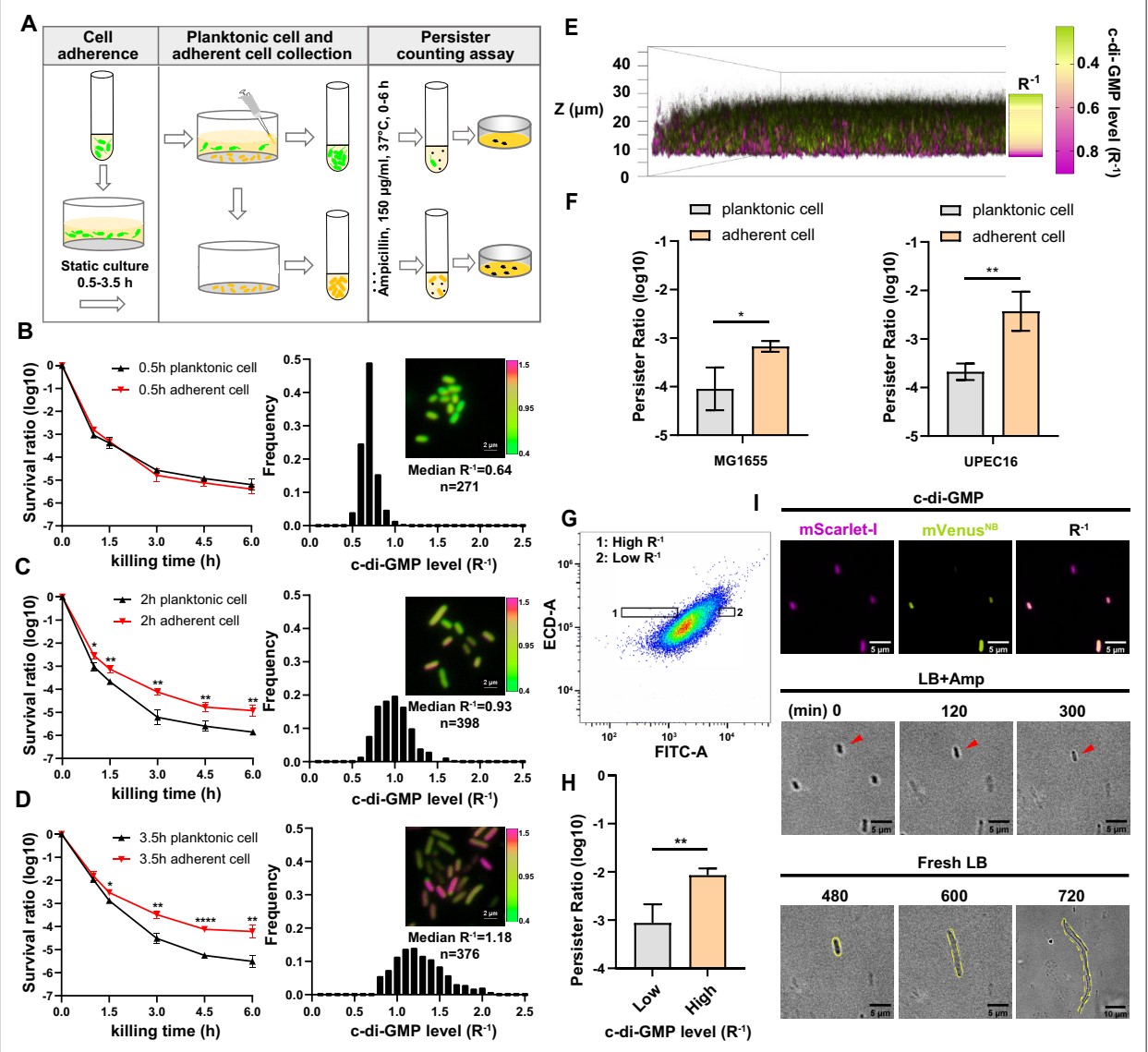

**Figure 1.** Cell adhesion promotes elevation of c-di-GMP levels and persister formation. (**A**) Schematic overview of the static biofilm system designed to investigate intrinsic factors influencing persister cell formation during the early stages of biofilm development. (**B–D**) Left panel: the biphasic killing curves of planktonic cell subpopulation and surface-adherent cell subpopulation from static biofilm system at the culture time of 0.5 hr (**B**), 2 hr (**C**) or 3.5 hr (**D**), respectively (n=3). Right panel: Frequency distribution of c-di-GMP levels ($R^{-1}$ score) within the surface-adherent subpopulation from the same static biofilm system and time points. Insert fluorescent images provide representative illustrations of c-di-GMP levels in adherent cells, measured using the c-di-GMP sensor system (***Figure 1—figure supplement 1B***) integrated into *E. coli* cells. Scale bar, 2 µm. $R^{-1}$ score was determined using the fluorescent intensity of mVenus$^{NB}$ and mScarlet-I in the system. (**E**) Spatial heterogeneity in c-di-GMP levels ($R^{-1}$ score) within a 24 hr static biofilm in MG1655 strain captured through confocal microscopy. (**F**) The persister ratio in planktonic cell subpopulation and surface-adherent cell subpopulation collected from the 24 hr static biofilms (ampicillin, 150 µg/ml, 3 hr at 37 °C with shaking at 220 rpm) (n=3). (**G–H**) The persister ratio (n=4) (**H**) in High-$R^{-1}$ or Low-$R^{-1}$ subpopulations sorted from the 24 hr static biofilms in MG1655 strain (ampicillin, 150 µg/ml, 3 hr at 37 °C with shaking at 220 rpm) (**G**). (**I**) Representative time-lapse images of the persister assay using cells with different $R^{-1}$ values. The MG1655 cells resuspended from 24 hr a static biofilm were imaged on a gel pad, and subsequently treated with LB broth containing 150 µg/ml ampicillin, imaged over 6 hr at 35 °C and then allowed to resuscitate in fresh LB. Scale bar, 5/10 µm, as indicated. Error bars represent standard deviations of biological replicates. Significance was ascertained by two-tailed Student's t test. Statistical significance is denoted as *p<0.05, **p<0.01, ***, p<0.001, ****, p<0.0001.

The online version of this article includes the following source data and figure supplement(s) for figure 1:

**Source data 1.** Related to *Figure 1*.

**Figure supplement 1.** A c-di-GMP biosensor was used in UPEC16 strain.

**Figure supplement 1—source data 1.** Related to *Figure 1—figure supplement 1*.

reduced fluorescence signal when the biosensor is bound to varying levels of c-di-GMP. Furthermore, a red fluorescent protein mScarlet-I is co-expressed under the regulation of the *proC* promoter (*Davis et al., 2011*), enabling the normalization of the system's expression level. Consequently, the inverse of the ratio between mVenus$^{NB}$-MrkH and mScarlet-I ($R^{-1}$) exhibits a direct correlation with the intracellular level of c-di-GMP, establishing a reliable indicator for monitoring c-di-GMP dynamics in UPEC16 (*Figure 1—figure supplement 1B*).

By monitoring the fluorescent intensity of the system at the single-cell level under a time-lapse microscope, we observed the dynamic activation of c-di-GMP during the process of cell adhesion. We imaged subpopulations on the surface at varied intervals. In the initial 30 min following static incubation in microtiter dishes, the adhered subpopulation displayed a sharply peaked $R^{-1}$ distribution with a low median of 0.64 (*Figure 1B*). As the attachment duration increased to 2 hr, the $R^{-1}$ distribution broadened, accompanied by a higher median of 0.93 (*Figure 1C*). Further extension to 3.5 hr resulted in a continued broadening of the $R^{-1}$ distribution, accompanied by an even higher median of 1.18 (*Figure 1D*). Meanwhile, the $R^{-1}$ distribution of the planktonic subpopulation from the same culture system and time points remained low, with a median of 0.31–0.35, repectively (*Figure 1—figure supplement 1C–E*). These findings replicate the observed positive correlation between c-di-GMP production and attachment duration. Moreover, the results suggest a positive association between c-di-GMP levels and the persister rate (*Figure 1B–D*).

Microcolonies emerged on the surface of microtiter dishes when the duration of static culture extended to 24 hr (*Figure 1E*). We employed confocal laser scanning microscopy to monitor spatial distribution of c-di-GMP signals within these microcolonies. This investigation revealed a stratified pattern of c-di-GMP levels across the cross-section of the static biofilms ($R^{-1}$, *Figure 1E*), highlighting the intricate regulatory role of this signaling molecule in shaping bacterial behavior and biofilm structure. Notably, cells with the highest c-di-GMP levels were surface attached, predominantly situated at the bottom layers of the biofilm (*Figure 1E*). To further investigate this observation, we isolated the surface attached cells and unattached cells in the biofilms for persister counting assay. The results demonstrate that this spatial distribution aligned with a subpopulation that exhibited the highest persister rate (*Figure 1F*). Furthermore, we resuspended the cells within the static biofilms and used FACS to isolate subpopulations with varying levels of c-di-GMP (*Figure 1G*). An antibiotic killing assay revealed that subpopulations with higher concentrations of c-di-GMP had a higher frequency of persister cells (*Figure 1H*). Time-lapse imaging also revealed that cells with high intracellular c-di-GMP levels survived antibiotic treatment and resuscitated to form microcolonies (*Figure 1I*). These findings collectively suggest that increased levels of cellular c-di-GMP play a crucial role in promoting persister formation.

## Heightened c-di-GMP levels promotes persister formation

To investigate the involvement of c-di-GMP in promoting the transition of persister phenotype in biofilms, we first constructed a UPEC16 strain with the ability to degrade c-di-GMP in a controlled manner. This was achieved through the overexpression of PdeH, an efficient c-di-GMP phosphodiesterase in *E. coli*, under the control of the araBAD promoter (pBAD::*pdeH*; *Pesavento et al., 2008*). Inducing PdeH expression before cell attachment disrupted cell adhesion, validating the efficacy of our system. Initiating with uninduced cells, we allowed them to adhere to a surface in a microliter dish (*Figure 2A*). After a 2 hr incubation, we observed robust cell adhesion similar to the vector control, reflected in comparable c-di-GMP $R^{-1}$ medians. Following this initial observation, we replaced the old medium with fresh medium containing arabinose (0.2%) for PdeH induction. After an additional 30 min of incubation, we noted a substantial decrease in c-di-GMP $R^{-1}$ levels in the PdeH overexpression strain (median of 0.75) compared to the vector control strain (median of 0.98; *Figure 2B–C*). Crucially, the biphasic killing curve showed significantly lower persister rates in adhered populations with PdeH overexpression compared to the vector control strain (*Figure 2D*). These findings suggest that c-di-GMP plays a pivotal role in promoting persister cell formation during cell-surface adhesion.

In parallel, we engineered a UPEC16 strain overexpressing DgcZ, a highly efficient diguanylate cyclase responsible for synthesizing c-di-GMP, under the araBAD promoter (pBAD::*dgcZ*). In a consistent approach, we allowed the cells to adhere to a surface in a microliter dish before induction (*Figure 2A*). After a 2 hr incubation, we observed cell adhesion comparable to the vector control, reflected in similar c-di-GMP $R^{-1}$ medians. Subsequently, we replaced the old medium with fresh

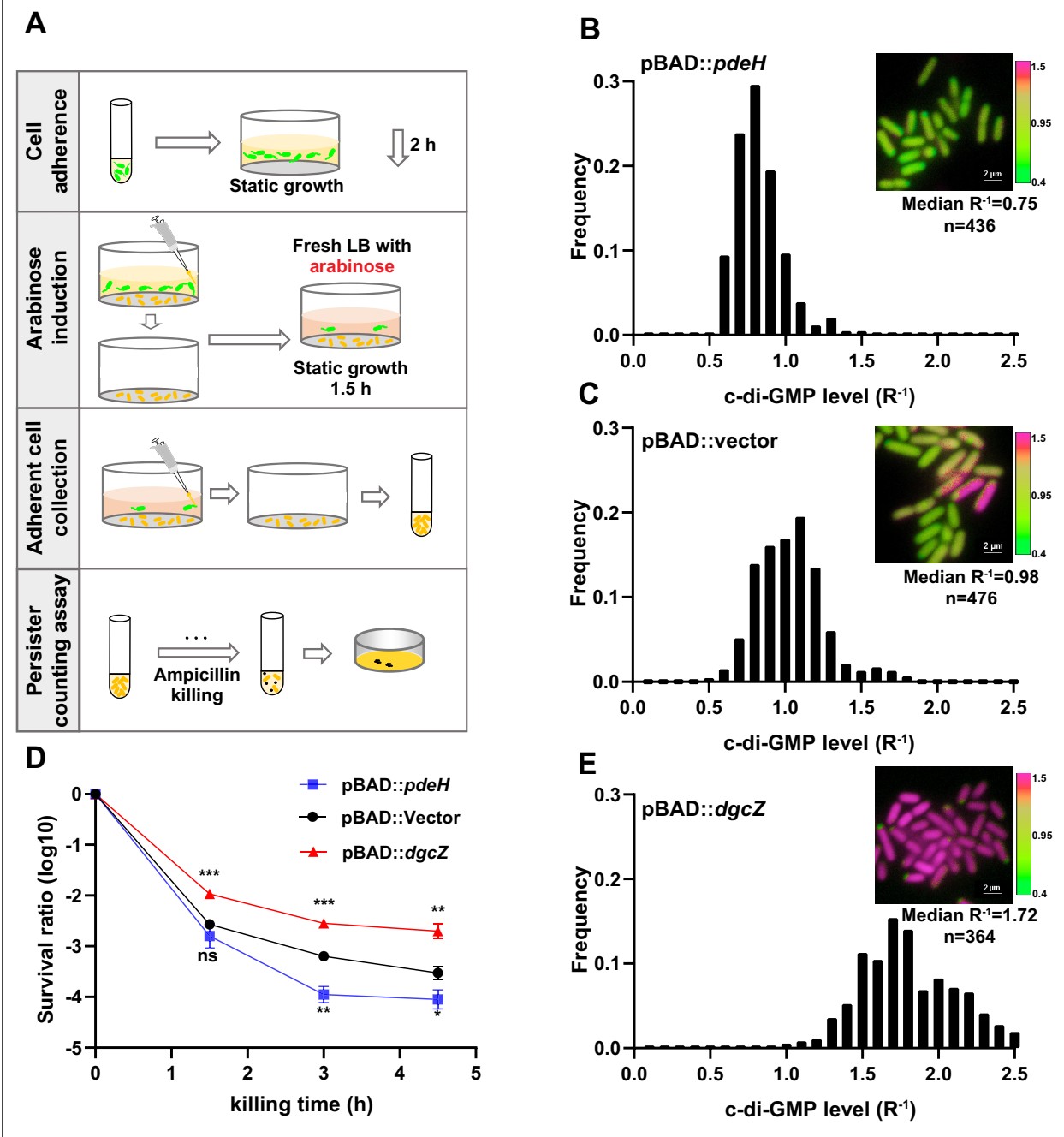

**Figure 2.** High c-di-GMP levels promote persister formation. (**A**) Schematic overview illustrating the manipulation of c-di-GMP levels in surface-adherent cell subpopulations for persister ratio determination. (**B–C, E**) Frequency distribution of c-di-GMP levels ($R^{-1}$ score) within the surface-adherent subpopulation from strains overexpressing either PdeH (pBAD::*pdeH*) (**B**) or DgcZ (pBAD::*dgcZ*) (**E**), along with a vector control (pBAD::vector) (**C**). Cells preparation followed the procedure depicted in (**A**). Insert fluorescent images provide representative illustrations of c-di-GMP levels in adherent cells, measured using the c-di-GMP sensor system (*Figure 1—figure supplement 1B*) integrated into different stains. Scale bar, 2 μm. (**D**) Biphasic killing curves of the surface-adherent subpopulations from (**A**) (n=3). Error bars represent standard deviations of biological replicates. Significance was ascertained by two-tailed Student's t test. Statistical significance is denoted as *p<0.05, **p<0.01, ***p<0.001.

The online version of this article includes the following source data for figure 2:

**Source data 1.** Related to *Figure 2*.

medium containing arabinose (0.2%) for DgcZ induction. Following an additional 30 min of incubation, we noted a substantial increase in c-di-GMP R$^{-1}$ levels in the DgcZ overexpression strain (median of 1.72) compared to the vector control strain (*Figure 2E*). Consistently, the biphasic killing curve indicated significantly higher persister rates in adhered populations with DgcZ overexpression compared to the vector control strain (*Figure 2D*). These findings reinforce the critical role of c-di-GMP in promoting persister cell formation during cell-surface adhesion.

## Toxin protein HipH acts downstream of c-di-GMP signaling in regulating persister formation in biofilms

Consistent with the observations in adherent cells, planktonic cells with DgcZ overexpression demonstrated a significant increase in the persister ratio (*Figure 3A*). Yet, cells overexpressing DgcZ-mutant (DgcZ-m, G206A, G207A, E208Q), which lacks diguanylate cyclase activity (*Boehm et al., 2009*; *Jonas et al., 2008*), did not manifest a parallel increase (*Figure 3A*). These results reinforce that elevated levels of c-di-GMP play a key role in promoting persister cell transition. To explore the mechanisms by which elevated c-di-GMP levels promote bacterial persister formation, we employed a systematic screening approach using the *E. coli* Keio Knockout Collection (KEIO collection; *Baba et al., 2006*). Given that the K-12 strain, the background strain of the KEIO collection, might not be ideal for cell adhesion and biofilm formation, we initially introduced the pBAD::*dgcZ* construct into individual mutant strains. This allowed us to establish a background characterized by increased c-di-GMP levels (*Figure 3B*). Subsequently, persister counting assays were performed by exposing exponentially growing transformed strains to the antibiotic ampicillin at a concentration of 150 µg/ml for a duration of 3 hr, followed by plating. This experimental design allowed us to assess the impact of upregulated c-di-GMP on persister cell formation within specific genetic backgrounds. Results show that knocking out the toxin gene *hipH* significantly reduces the frequency of persister formation in the presence of high levels of c-di-GMP (*Figure 3C*). Meanwhile, the knockout of *hipH* alone does not lead to a significant alteration in persister frequency during the exponentially growing phase. These observations suggest that HipH acts downstream of c-di-GMP signaling in the process of persister formation and antibiotic tolerance. Considering HipH belongs to the family of HipA-like Ser/Thr kinases, we investigated the impact of knocking out *hipA*. In contrast, knocking out *hipA* had little effect on the frequency of persister formation with elevated c-di-GMP levels (*Figure 3C*). This observation suggests that HipH plays a distinct role from HipA in inducing persister formation.

Given that c-di-GMP exhibits upregulation in cell-surface adhesion, we conducted a *hipH* knockout in UPEC16 strain (UPEC16-Δ*hipH*) and evaluated its influence on persister formation in biofilms (*Figure 3D*). In in vitro biofilms, we observed a substantial decrease in persister levels following *hipH* knockout, contrasting with the wild-type strain that consistently displayed notably high levels of persister cells (*Figure 3E*). These results suggest that c-di-GMP regulated HipH toxicity plays a crucial role in persister formation of biofilms. For in vivo biofilm formation, we introduced both the wild-type and UPEC16-Δ*hipH* strains, each at a concentration of 2*10^8 CFU, into the mouse bladders through transurethral catheterization (*Pang et al., 2022*; *Figure 3D*). Following 24 hr duration of infection, both strains exhibited comparable levels of intracellular bacterial communities (IBC) within the epithelial cells lining the bladder lumen (*Figure 3—figure supplement 1A–B*). Subsequently, the infected mice were sacrificed, and their bladders were isolated for colony counting assays and antibiotic killing assays. Consistent with the findings from in vitro biofilms, the wild-type UPEC16 strain displayed a notably high rate of persister formation. In contrast, the UPEC16-Δ*hipH* strain exhibited a substantial reduction in persister cell frequency under the same conditions (*Figure 3F*). The impact of HipH knockout on persister formation is consistently observed in both in vitro and in vivo settings associated with biofilms, but not in planktonic populations (*Figure 3C*, left panel). These results reinforce HipH toxicity is under the regulation of c-di-GMP, and emphasize the significance of HipH in shaping the persistence and antibiotic tolerance characteristics of biofilms.

We observed a significant elevation in *hipH* gene expression upon the overexpression of DgcZ (pBAD::*dgcZ*) (*Figure 3G–H*). Conversely, the strain overexpressing DgcZ-m, which is unable to increase c-di-GMP levels after induction, exhibit no activation of *hipH* gene expression (*Figure 3G*). Furthermore, we labeled HipH with BFP at the C-terminal end and placed it under the *hipH* native promoter in the UPEC16 strain with c-di-GMP sensing system. Fluorescent imaging revealed a positive correlation between HipH and c-di-GMP expression at the single-cell level (*Figure 3—figure*

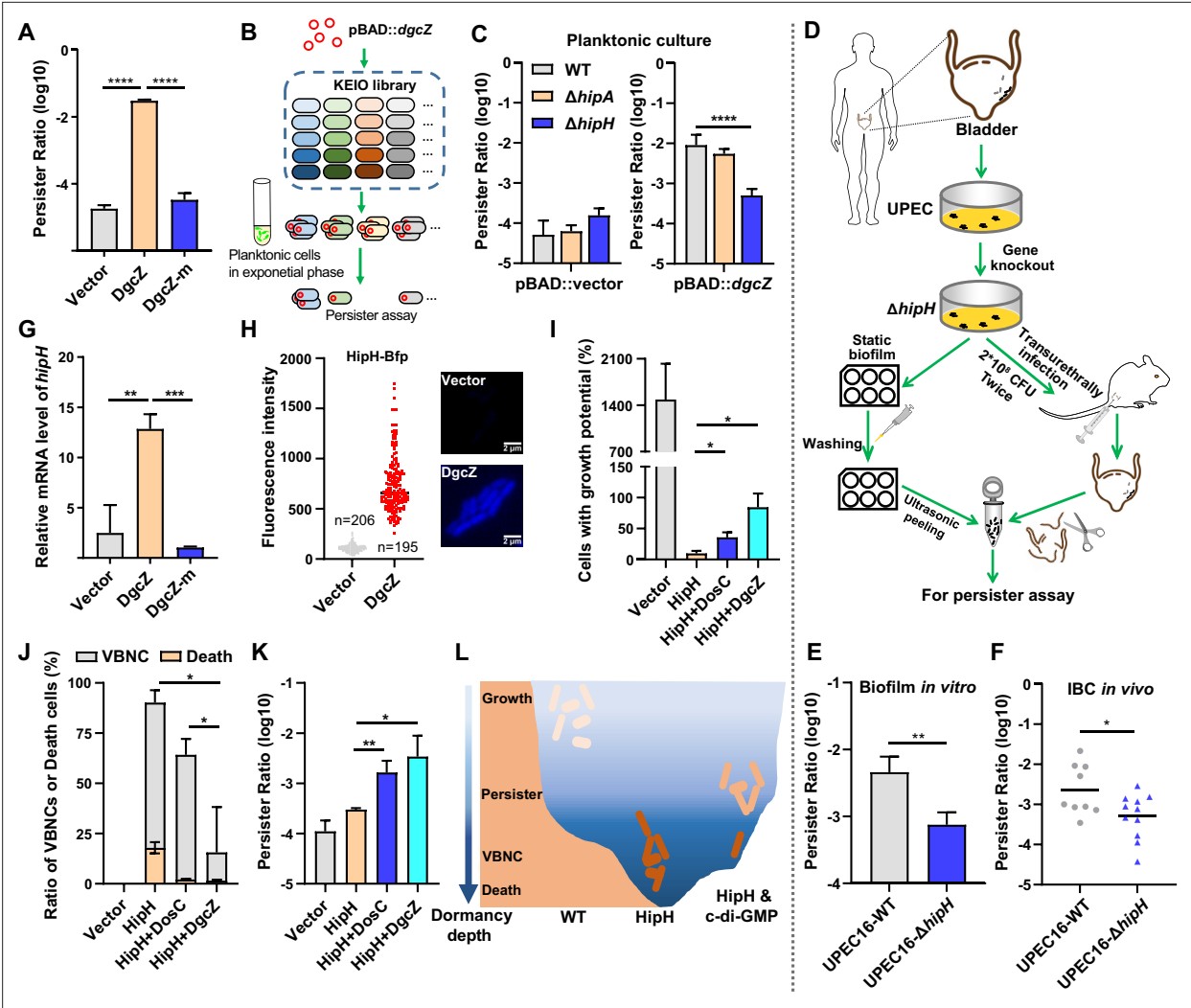

**Figure 3.** HipH and c-di-GMP as a TA-like system. (**A**) Determination of persister ratios in MG1655 strains overexpressing DgcZ or DgcZ-m (G206A, G207A, E208Q), with 0.002% arabinose induction for 2 hr in the pBAD plasmid. Persister counting was performed following antibiotic killing (ampicillin, 150 µg/ml, 3 hr at 37 °C with shaking at 220 rpm) (n=3). (**B**) Schematic overview illustrating the screening of downstream targets of c-di-GMP by using *E. coli* Keio Knockout Collection, the screen was conducted in the presence of high levels of c-di-GMP by inducing the expression of DgcZ with 0.002% arabinose for 2 hr. (**C**) Identification of HipH as the downstream target of c-di-GMP in persister formation in Keio strains. Left panel: persister assay performed without high levels of c-di-GMP (pBAD-vector); Right panel: persister assay performed in the presence of high levels of c-di-GMP (pBAD-*dgcZ*) (ampicillin, 150 µg/ml, 3 hr at 37 °C with shaking at 220 rpm) (n=9). (**D**) Schematic overview of clinical UPEC16 isolation, genetic modification, and in vitro or in vivo biofilm culture. (**E**) Antibiotic killing (ampicillin, 150 µg/ml, 3 hr at 37 °C with shaking at 220 rpm) and persister counting assay for in vitro biofilm cells (n=3). (**F**) Antibiotic killing (ampicillin, 150 µg/ml, 3 hr at 37 °C with shaking at 220 rpm) and persister counting assay for in vivo IBC cells (n=9 for WT, n=11 for Δ*hipH*). (**G**) Relative expression level of *hipH* in MG1655 strains under different levels of cellular c-di-GMP, determined by RT-qPCR and normalized to the levels of 16 S rRNA. DgcZ and DgcZ-m were induced by 0.002% arabinose for 2 hr in the pBAD plasmid (n=3). (**H**) Fluorescence intensity of HipH fused to BFP in MG1655 strains under varying levels of cellular c-di-GMP. DgcZ was induced by 0.002% arabinose for 2 hr using the pBAD plasmid. The inserted fluorescent images provide representative illustrations of HipH-BFP. Scale bar, 2 µm. (**I–K**) Detection of the ratio of cells with growth potential (**I**), VBNCs or death cells (**J**) and persisters (**K**) in MG1655 strains overexpressing HipH alone, or overexpressing HipH and DosC together, or overexpressing HipH and DgcZ together, induced by 0.002% arabinose for 4 hr. Persister counting performed followed antibiotic killing (ampicillin, 150 µg/ml, 3 hr at 37 °C with shaking at 220 rpm) (n=3). (**L**) Schematic representation of the dormancy depth controlled by HipH and c-di-GMP, and the relationship with cell state. Error bars represent standard deviations of biological replicates. Significance was ascertained by two-tailed Student's t test (ACEFGHK) or Two-way ANOVA (**I–J**). Statistical significance is denoted as *p<0.05, **p<0.01, ***, p<0.001; ****, p<0.0001.

The online version of this article includes the following source data and figure supplement(s) for figure 3:

**Source data 1.** Related to *Figure 3*.

**Figure supplement 1.** UPEC16 infection of mouse bladder.

**Figure supplement 1—source data 1.** Related to *Figure 3—figure supplement 1*.

*Figure 3 continued on next page*

*Figure 3 continued*

**Figure supplement 2.** The correlation between the levels of c-di-GMP and HipH.

**Figure supplement 2—source data 1.** Related to *Figure 3—figure supplement 2*.

supplement 2A), supported by a correlation coefficient of 0.694 (*Figure 3—figure supplement 2B*). Additionally, we engineered an *E. coli* strain in which *hipH* promoter is replaced by the constitutive promoter p-rhaB (*Figure 3—figure supplement 2C*). In this strain, *hipH* expression remains unaffected by changes in c-di-GMP levels (*Figure 3—figure supplement 2D*), indicating that regulation occurs at the promoter region. Furthermore, the persister ratio was comparable to that of the Δ*hipH* strain but significantly lower than that of wild-type strain under high c-di-GMP levels (*Figure 3—figure supplement 2E*). These results together suggest that c-di-GMP exerts a transcriptional regulatory role in controlling the expression of HipH.

## c-di-GMP and HipH form a TA-like module

We then independently overexpressed HipH (pBAD::*hipH*) to observe its impact on cell growth. The results demonstrated that HipH overexpression substantially inhibit cell growth, with only 9.8 ± 3.4% of cells retaining growth potential (*Figure 3I*). Further analysis of the growth-inhibited cells revealed a predominant presence of Viable but Non-Culturable (VBNC) cells (*Ayrapetyan et al., 2018*; *Liu et al., 2023*; *Pan and Ren, 2022*), constituting 72 ± 5% of the total cell population, alongside a significant proportion of dead cells, accounting for 17.9 ± 2.3% of the total cells (*Figure 3J*). This evidence suggests that HipH functions as a potent toxin capable of inhibiting cell growth or even inducing cell death. Interestingly, despite the observed decrease in growth potential, the persister rate remains low as that of wild-type strains (*Figure 3K*). This phenomenon is interpreted as cells entering a state of deep dormancy with reduced resuscitation capabilities (*Figure 3L*).

To test the role of c-di-GMP as an antitoxin in alleviating the toxic effect of HipH, we elevated intracellular c-di-GMP levels by overexpressing DosC, a moderately efficient diguanylate cyclase, or DgcZ, a highly efficient diguanylate cyclase, in the strain overexpressing HipH. This manipulation resulted in a 2.72-fold and 4.22-fold increase in the mean of c-di-GMP $R^{-1}$ score, respectively (*Figure 3—figure supplement 2F*). Comparing to the strain overexpressing HipH alone, those strains co-overexpressing HipH and c-di-GMP exhibited a significantly recovered growth potential (*Figure 3I*). Specifically, 36 ± 6.7% of cells retained growth potential in the strain expressing HipH and DosC, while approximately 85 ± 18% retained growth potential in the strain expressing HipH and DgcZ. Further analyzing the portion of growth-inhibited cells revealed that the strain overexpressing HipH and DosC exhibited a decreased rate of VBNC cells (62 ± 6% of the total cell population) and dead cells (2.1 ± 0.2% of the total cell population; *Figure 3J*). The strain overexpressing HipH and DgcZ showed an even lower rate of VBNCs (14 ± 17% of the total cell population) and dead cells (1.5 ± 0.4% of the total cell population; *Figure 3J*). These results support the assumption that c-di-GMP acts as an effective antitoxin, enabling HipH-toxified cells to recover their growth potential. Consistently, increased rates of persister cells were observed, with $10^{-2.8 \pm 0.19}$ in the strain overexpressing HipH and DosC, and $10^{-2.5 \pm 0.34}$ in the strain overexpressing HipH and DgcZ (*Figure 3K*). These results collectively indicate that c-di-GMP functions as an antitoxin in alleviating HipH's harmful effects, steering cells from deep dormancy depth to a shallower dormancy state with enhanced resuscitation capabilities, thereby promoting persister cell formation (*Figure 3L*).

## Hiph Is a Deoxyribonuclease Inducing Genome Instability

*hipH* stands out as an atypical toxin gene due to its lack of a genetically linked antitoxin cognate, existing as a solitary gene unlike the typical toxin-antitoxin gene pairs found within the same operon (*Jurėnas et al., 2022*; *Maeda et al., 2017*). Previous studies believed it a homolog of the toxin protein HipA, functioning as a Ser/Thr protein kinase involved in regulating ribosome assembly and cell metabolism (*Gratani et al., 2023*). However, our findings challenge this prevailing belief. The truncated mutant of HipH, lacking the kinase catalytic motif (Δ5, amino acid 340–372), demonstrated a comparable ability to inhibit cell growth and induced VBNCs compared to the intact HipH (*Figure 4A–C*). These results suggest the involvement of another domain in HipH that is responsible for its toxic activity.

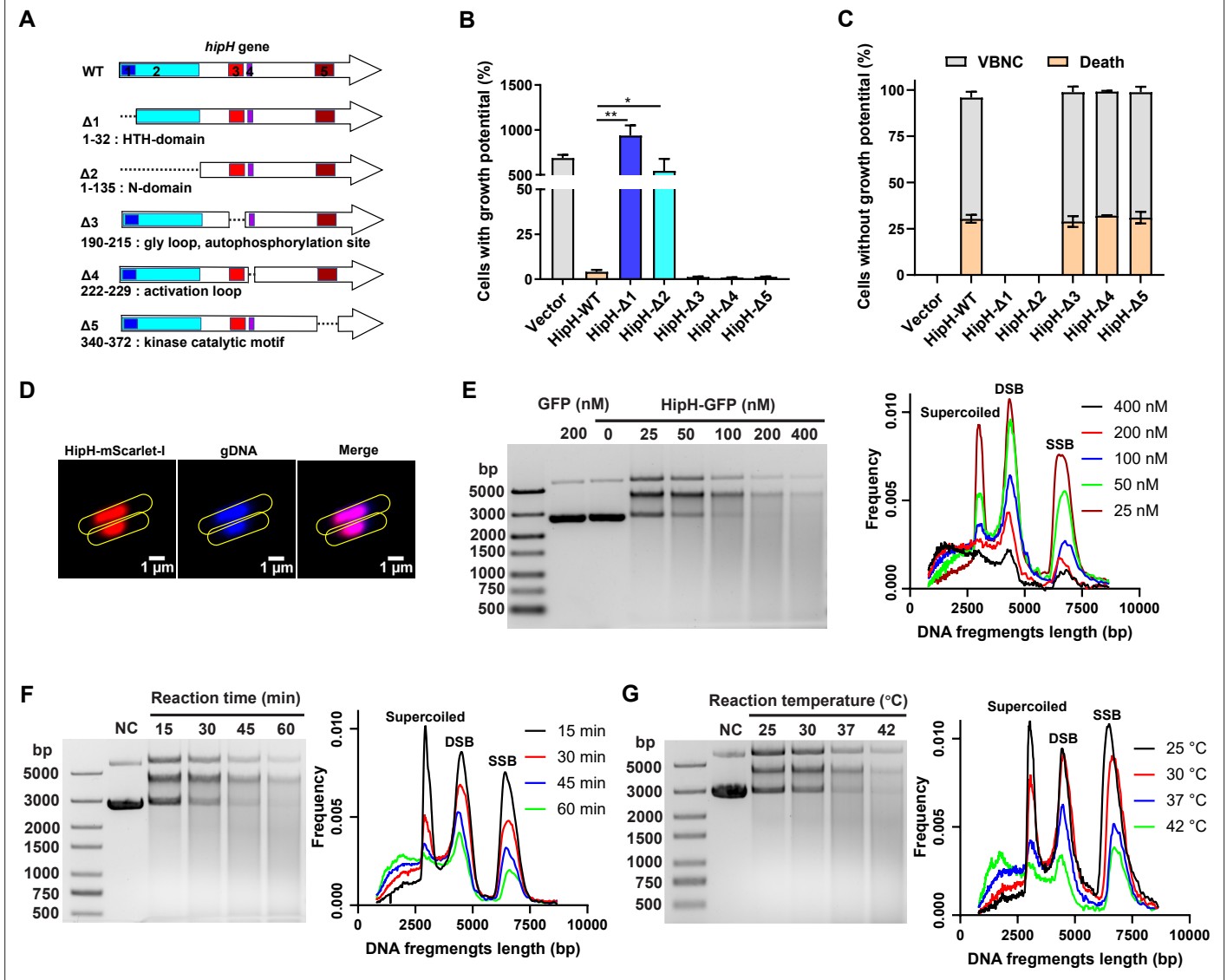

**Figure 4.** HipH is a deoxyribonuclease exerting genotoxic effects. (**A**) Structure analysis of HipH protein and Schematic representation of truncated mutants illustrating key structural domains. (**B–C**) Detection of the ratio of cells with growth potential (**B**) and VBNCs or death cells (**C**) in MG1655 strains overexpressing different truncated HipH mutants, induced by 0.002% arabinose for 3 hr (n=3). (**D**) Subcellular localization of HipH in MG1655 strain. HipH was fused with mScarlet-I and induced by 0.002% arabinose for 2 hr. The gDNA was stained with Hoechst 33258 for 15 min at room temperature. Scale bar, 1 μm. (**E–G**) Left panel: in vitro DNA cleavage assay using 500 ng supercoiled plasmid DNA, (**E**) with varying concentrations of HipH-GFP (0, 25, 50, 100, 200, or 400 nM), with purified GFP protein as control; (**F**) with different durations of reaction time (15, 30, 45, and 60 min); (**G**) with different reaction temperature (25, 30, 37, and 42°C). If not specified, 200 nM of HipH-GFP was added to each reaction. 'NC', negative control, indicates supercoiled plasmid without addition of HipH-GFP. Molecular weight markers from top to bottom are 5000, 3000, 2000, 1500, 1000, 750, 500 (bp). Right panel: illustration of the relationship between the length of DNA fragments and their ratio to total DNA, as indicated in the corresponding upper panel. SSB, single-strand break; DSB, double strand break. Error bars represent standard deviations of biological replicates. Significance was ascertained by Two-way ANOVA. Statistical significance is denoted as *p<0.05, **p<0.01.

The online version of this article includes the following source data and figure supplement(s) for figure 4:

**Source data 1.** Related to *Figure 4*.

**Figure supplement 1.** HipH induced genome instability.

**Figure supplement 1—source data 1.** Related to *Figure 4—figure supplement 1*.

Structure analysis revealed that, in addition to its kinase catalytic motif (amino acid 340–372), HipH also contains a DNA binding HTH domain (amino acid 11–32), a gly loop and autophosphorylation site (amino acid 190–215), and an activation loop (amino acid 222–229; *Figure 4A*). To identify the specific domain responsible for its toxicity, we generated various truncated mutants and evaluated their capacity in inhibiting cell growth. Our results showed that overexpression of mutants lacking the DNA binding domain HTH (Δ1 and Δ2) did not inhibit cell growth or induce VBNCs or dead cells (*Figure 4B–C*). This indicates that the DNA binding domain HTH is indispensable for the observed toxicity of HipH. Consistent with these results, fluorescent microscopy confirmed the nucleoid localization of wild-type HipH (*Figure 4D*), indicating that the primary function of HipH is related to DNA interaction.

We observed the elongated filamentous shape induced by HipH overexpression in cells (*Figure 4— figure supplement 1A*), which is consistent with previous findings (*Gratani et al., 2023*). We interpret this as a potential characteristic associated with DNA damage. Furthermore, considering the results of the truncated mutation assay (*Figure 4A–C*), we hypothesize that HipH plays a role in inducing DNA damage. To investigate this hypothesis, we first examined whether HipH overexpression induces the SOS response, a bacterial defense mechanism against DNA damage involving the expression of various genes. qPCR analysis revealed a significant upregulation of SOS response genes, such as *recA*, *sulA*, and *tisB*, in cells overexpressing HipH (*Figure 4—figure supplement 1B*). Notably, the near abolition of cell growth potential upon knockout of *recA*, a crucial protein essential for DNA damage repair, in cells overexpressing HipH further supports the conclusion that HipH induces DNA damage (*Figure 4—figure supplement 1C*).

To investigate the specific type of DNA damage caused by HipH, we conducted in vitro DNA-cleaving assays (*Hockings and Maxwell, 2002*; *Maki et al., 1996*) using purified HipH. To address its insolubility, we engineered a fusion construct with a high solubility protein, GFP, at the C-terminal end to facilitate protein purification. The results demonstrated HipH's capability to cleave supercoiled plasmids, leading to the production of double-strand breaks (DSBs), as evidenced by prominent smears observed on agarose gels (*Figure 4E–G*). Further analysis revealed a positive correlation between HipH's DNA cleavage capacity and its protein concentration, temperature, and incubation time (*Figure 4E–G*). These findings collectively indicate that HipH cleaves DNA and produces DSBs. The overexpression of RecG, a key player in the repair of DSBs in DNA (*Whitby et al., 1994*), successfully rescued growth-deficient cells resulting from HipH overexpression (*Figure 4—figure supplement 1D*), providing additional evidence that HipH induces DSBs in DNA. These compelling results demonstrate that HipH functions as a deoxyribonuclease, exerting genotoxic effects on genomic DNA integrity.

## c-di-GMP acts as an antitoxin to alleviate HipH genotoxicity

To investigate whether c-di-GMP plays as an antitoxin to mitigate HipH genotoxicity, we first explored the potential of direct interaction between c-di-GMP and HipH. A biotinylated c-di-GMP pull-down assay (*Figure 5A*; *Chambers and Sauer, 2017*), conducted from both cell lysate and purified protein, revealed a physical binding between c-di-GMP and HipH (*Figure 5B–D*, *Figure 5—figure supplement 1A–B*), confirming their direct interaction. The dissociation constant ($K_D$), calculated based on the concentration of c-di-GMP at which half-maximal binding of HipH occurs, was determined as 7.6 µM (*Figure 5B*, bottom panel). Additionally, c-di-GMP successfully displaced HipH-GFP-His from HipH-GFP-His/biotin-c-di-GMP complexes, while other nucleotides such as cAMP and GTP did not exhibit the same effect (*Figure 5E*). This underscores the specificity of the binding between c-di-GMP and HipH.

Subsequently, we introduced c-di-GMP into the HipH DNA-cleavage assay system, and notably, its addition exerted a significant suppressive effect on the cleavage activity of HipH. This effect was observed for both relaxed and supercoiled plasmids, as evidenced by a reduction of smears and double-strand break (DSB) bands on agarose gels (*Figure 5F–I*). It is noteworthy that other nucleotides, including cAMP and GTP, did not demonstrate the same suppressive effect on HipH's cleavage activity (*Figure 5—figure supplement 1C*). Furthermore, it has been reported that HipB serves as an antitoxin for HipH (*Gratani et al., 2023*; *Maeda et al., 2017*). However, when we tested the potential impact of HipB on HipH's DNase activity by purifying and introducing HipB into our cleavage system, we found that HipB failed to inhibit HipH's DNase activity (*Figure 5—figure supplement 1C*). These

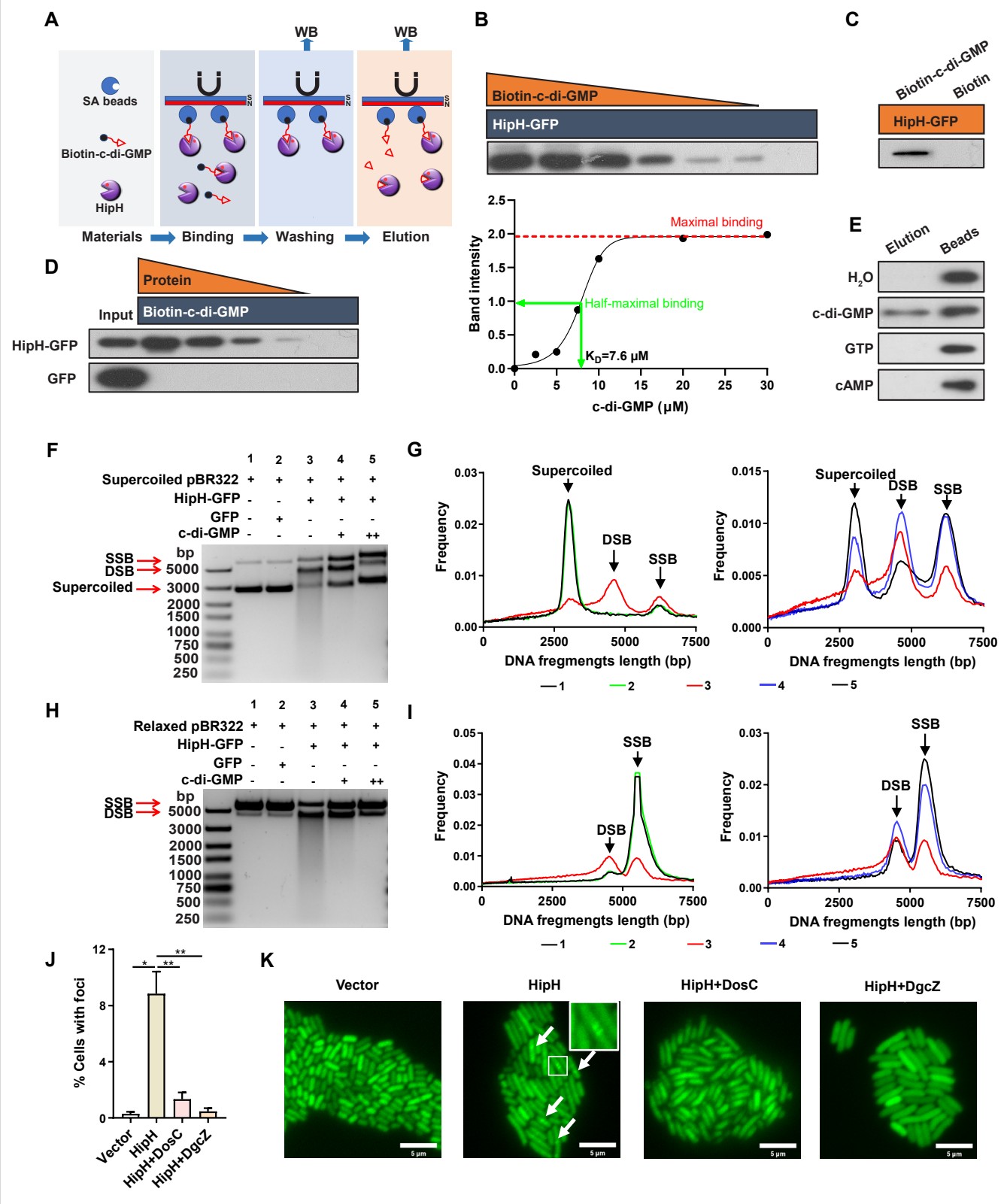

**Figure 5.** c-di-GMP acts as an antitoxin to repress HipH DNA cleavage activity. (**A**) Schematic representation of the biotin-c-di-GMP pull-down assay for HipH protein. (**B–E**) Identification of the interaction between c-di-GMP and HipH using the biotinylated c-di-GMP pull-down assay. (**B**) Upper panel: 1.5 µM purified HipH-GFP-His and biotin-c-di-GMP were added in gradient concentrations (30, 20, 10, 7.5, 5, 2.5, 0 µM). Bottom panel: the intensity of bound protein, obtained by analyzing the immunoblot shown in the upper panel using ImageJ, relative to the concentrations of c-di-GMP used in the

*Figure 5 continued on next page*

*Figure 5 continued*

pull-down assays. (**C**) 0.3 μM purified HipH-GFP-His and 20 μM biotin-c-di-GMP or biotin were added. (**D**) Purified HipH-GFP-His or GFP-His was added in gradient concentrations (1.5, 0.75, 0.375, 0.19, 0 μM), and 20 μM biotin-c-di-GMP was added. 0.5 μg protein was used as the input sample. (**E**) 2 mM c-di-GMP, cAMP, or GTP were used to elute HipH-GFP-His from SA beads, as well as HipH-GFP-His/biotin-c-di-GMP complexes prepared from the pull-down reaction mix containing 1.5 μM purified HipH-GFP-His and 20 μM biotin-c-di-GMP. (**F–I**) In vitro DNA cleavage assay using 500 ng supercoiled plasmid DNA (**F–G**) or relaxed plasmid DNA (**H–I**).'+' indicated 1.67 mM c-di-GMP and '++' indicated 5 mM c-di-GMP. 200 nM HipH-GFP-His or GFP-His was used in Cleavage reaction at 37 °C for 1 hr. Molecular weight markers from top to bottom are 5000, 3000, 2000, 1500, 1000, 750, 500, 250 (bp). (**G**) and (**I**) represent the relationship between the length of DNA fragments and their ratio to total DNA, as shown in B and D, respectively. SSB: single-strand break (relaxed plasmid DNA); DSB: double-strand break. (**J**) Statistic analysis of the ratio of cells with Gam-GFP foci in different MG1655 strains: overexpressing HipH alone, or HipH with DosC, or HipH with DgcZ (n=3). (**K**) Representative images of cells expressing Gam-GFP in the aforementioned different strains. Scale bar, 5 μm. Error bars represent standard deviations of biological replicates. Significance was ascertained by Two-way ANOVA. Statistical significance is denoted as *p<0.05, **p<0.01.

The online version of this article includes the following source data and figure supplement(s) for figure 5:

**Source data 1.** PDF containing original scans of the dot blot related to *Figure 5B–E*.

**Source data 2.** Related to *Figure 5*.

**Figure supplement 1.** c-di-GMP binding inhibits HipH DNA cleavage activity.

**Figure supplement 1—source data 1.** PDF containing original scans of the dot blot related to *Figure 5—figure supplement 1A, B*.

**Figure supplement 1—source data 2.** Related to *Figure 5—figure supplement 1*.

findings suggest that c-di-GMP specifically interacts with HipH, leading to the inhibition of its DNase activity.

To monitor DNA breakage in live bacterial cells with varying stoichiometry between HipH and c-di-GMP, we utilized Gam-GFP, a double-strand DNA end-binding protein of bacteriophage Mu that forms foci at DSBs in living cells (*Shee et al., 2013*). Microscopic examination revealed that cells over-expressing HipH alone displayed Gam-GFP foci in 8.85 ± 1.28% of cases, while only 0.30 ± 0.12% of wild-type cells demonstrated such foci (*Figure 5J–K*). Further investigations showed that elevation of intracellular c-di-GMP levels resulted in a significant reduction in the percentage of cells with Gam-GFP foci, with DosC addition decreasing the occurrence to 1.34 ± 0.39% and DgcZ to 0.46 ± 0.19% (*Figure 5J–K*). These in vitro and in vivo findings conclusively demonstrate that c-di-GMP acts as an antitoxin for HipH, effectively suppressing its genotoxic effects.

## Discussion

In the intricate biofilm dynamics, the origin of persister formation has largely been ascribed to the constraints imposed by the dense structures, such as limited nutrients, oxygen, and antibiotics penetration, or quorum sensing. Our investigation reveals that cell adhesion, a pivotal event initiating biofilm development, emerges as a trigger for persister formation. This concept introduces a layer of complexity that reshapes our understanding of drug tolerance in biofilms. Beyond the traditional emphasis on dense biofilm structures, the scope should now encompass single-layer cells on the surface. Such a deviation from conventional perspectives highlights the necessity for a more nuanced and comprehensive approach to address the challenges of antibiotic efficacy within biofilm communities.

### Mechanism linking surface adhesion to persister cell formation in bacterial biofilms

This study provides novel insights into the relationship between bacterial adhesion to surfaces and the subsequent increase in persister cell formation, which has not been explicitly detailed in previous literature. While existing research has established that biofilms typically harbor higher numbers of persister cells, this investigation not only corroborates that finding but also elucidates the mechanisms through which surface adhesion contributes to this phenomenon. Past studies have predominantly focused on the general characteristics of persister cells and their role in biofilm resilience and anti-biotic tolerance without specifically addressing the mechanistic link between adhesion and persister formation (*Pan et al., 2023*; *Patel et al., 2022*). For instance, previous work has shown that surface attachment leads to changes in metabolic activity and signaling pathways within bacterial cells, which

could promote persistence, but it has not definitively established a causal relationship between adhesion and increased persister formation. Our study highlights that the elevation of c-di-GMP levels after surface adhesion triggers a cascade of physiological changes that significantly enhance the formation of persister cells. In particular, we report that adhesion-induced signaling pathways promote dormancy and tolerance to antibiotics, marking an important advancement from the previous understanding that treated persister cells might arise from random phenotypic variation during biofilm development.

## Noncanonical toxin-antitoxin systems involving with small metabolic molecules?

c-di-GMP emerges as a pivotal secondary messenger in the intricate orchestration of bacterial lifestyles, playing a key role in mediating cell-surface adhesion and exerting influence over the transition from planktonic to sessile existence. Our comprehensive exploration, employing a spectrum of techniques, brings to light a significant revelation: heightened levels of c-di-GMP in cells drive the formation of persisters. Notably, c-di-GMP functions as a noncanonical antitoxin, regulating both the expression and genotoxicity of the toxin protein HipH. This small molecule, composed of just two guanine bases, counteracts the toxin's effects.

Another intriguing parallel to our findings can be drawn from the ppGpp-SpoT system, which exemplifies how small metabolic molecules can function in a manner reminiscent of traditional toxin-antitoxin (TA) systems (**Amato et al., 2013**). At the core of this system is the synthesis and hydrolysis of ppGpp (guanosine 5'-diphosphate 3'-diphosphate), modulating gene expression for stress adaptation. In this TA-like paradigm, ppGpp assumes the role of the 'toxin', while SpoT functions as the 'antitoxin'. SpoT exhibits dual functionality, both synthesizing and degrading ppGpp in response to various stresses, thereby regulating the levels of this potent signaling molecule.

Both systems play crucial roles in bacterial stress adaptation and persistence regulation. The ppGpp-SpoT system employs ppGpp as a toxin-like molecule coordinating various physiological processes. In contrast, the HipH-c-di-GMP interaction involves c-di-GMP as an antitoxin-like molecule influencing persister cell formation by regulating HipH, a genome instability-inducing toxin. These systems expand the conceptual framework of TA systems. While traditional TA systems consist of genetically linked gene pairs encoding a toxin and its neutralizing antitoxin (**Jurėnas et al., 2022**; **Qiu et al., 2022**; **Srivastava et al., 2021**), these new systems introduce additional complexities and nuances to

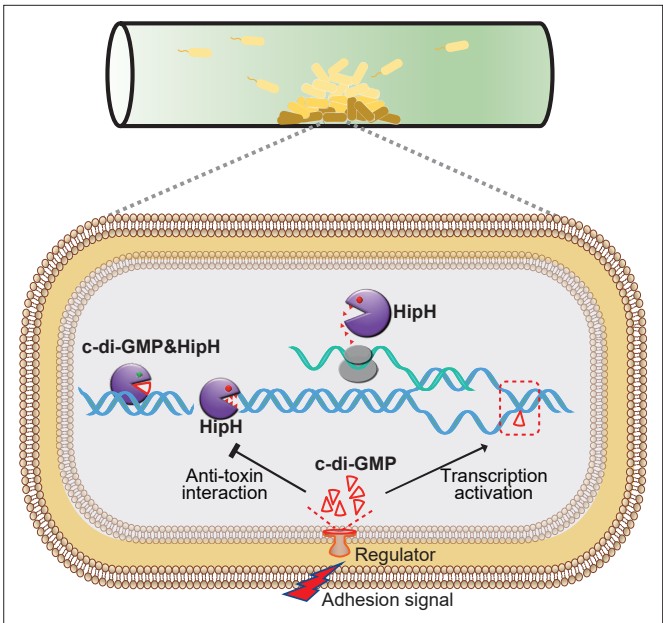

**Figure 6.** Schematic overview of the mechanism underlying c-di-GMP induced persister formation in biofilms. In biofilms, cell-surface adhesion triggers an increase of intracellular c-di-GMP levels. Elevated intracellular c-di-GMP levels inducing the expression of HipH. The HipH protein is a deoxyribonuclease which induces DNA double-strand break. Acting as an antitoxin, c-di-GMP mitigates the genotoxicity of HipH.

our understanding of bacterial regulatory mechanisms. The involvement of metabolites like ppGpp and c-di-GMP in TA-like systems demonstrates the intricate interplay between cellular metabolism and stress response pathways. This integration allows bacteria to fine-tune their responses to environmental challenges with remarkable precision. Moreover, these systems underscore the evolutionary adaptability of bacteria, showcasing how they have co-opted metabolic molecules into sophisticated regulatory networks.

### Novel functions of toxin protein HipH

HipH emerges as an intriguing toxin within the bacterial realm, standing apart due to its solitary existence without a genetically linked antitoxin cognate—an uncommon trait in known toxin-antitoxin systems. Previous studies have identified HipH as a Ser/Thr protein kinase, sharing homology with the well-characterized toxin HipA. These studies demonstrated its primary protein substrates to be the ribosomal protein RpmE (L31) and the carbon storage regulator CsrA (*Gratani et al., 2023*). While the overproduction of HipH was observed to impact bacterial DNA segregation and cell division, the underlying mechanism remained unexplained. In this study, we have uncovered that HipH primarily functions as a potent DNase, providing an explanation for the observed phenomenon. Our findings further reveal that that HipH, as a potent DNase, induces a deep cell dormancy depth and elevated rates of VBNC cells and dead cells, rather than persister cells with the capacity of resuscitation. Significantly, our investigation further sheds light on the regulation of HipH genotoxicity by c-di-GMP in a dual manner (*Figure 6*). Firstly, high intracellular levels of c-di-GMP induce HipH expression. Subsequently, c-di-GMP acts as an antitoxin, mitigating the potent toxicity of HipH. This, in turn, maintains cells with an adequate dormancy depth, facilitating the formation of persister cells and conferring antibiotic tolerance. It is intriguing to delve deeper into research exploring the domain-function relationship in HipH. However, our efforts have often encountered challenges, as mutations can readily induce protein aggregates, impeding thorough analysis. Achieving a comprehensive understanding of this domain-function relationship necessitates the development of innovative methodologies and approaches.

Taken together, we reveal a sophisticated regulatory circuit underlying bacterial genome instability and antibiotic persistence in biofilms. This regulated genome instability can have significant implications for adaptive potential of bacterial populations, shaping their ability to persist, evolve, and develop resistance mechanisms. By targeting this c-di-GMP regulated genotoxicity of toxin protein HipH, we successfully inhibited drug tolerance in UPEC induced biofilms. These findings provide potential therapeutic strategies for combating chronic and relapsing infections associated with biofilms.

## Materials and methods

### Bacterial strains and growth conditions

Bacterial strains used in this study included *Escherichia coli* strains MG1655, BW25113, UPEC16, and the Keio library Knockout Collection (Key resources table). UPEC16 was isolated from a urine sample (#16) obtained from a local urinary tract infection (UTI) patient at the First Affiliated Hospital of Kunming Medical University, following the guidelines of the Committee on Human Subject Research and Ethics (2022-L-203 and CHSRE2022030). The genome sequence has been deposited in the National Genomics Data Center under the accession number 'GWHERGI00000000'. All strains were cultured in Luria Broth (LB) medium, with routine growth condition at 37 °C and 220 rpm. To support plasmid maintenance, media were supplemented with the corresponding antibiotics as needed: chloramphenicol (25 µg/ml), kanamycin (20 µg/ml), gentamicin (20 µg/ml), and ampicillin (100 µg/ml). For inducible expression experiments, the medium was supplemented with induces such as 0.002%–0.2% arabinose for the Arabinose-induction system, 1 mM IPTG for the Lac-induction system, and 1 µg/ml anhydrotetracycline for the Tet-induction system.

### Strains construction

#### Construction of target DNA knockout

The generation of target DNA knockout strains involved the use of $\lambda$-red mediated gene replacement (*Datsenko and Wanner, 2000*). In this procedure, a 500 bp upstream region of the target DNA sequence, a resistance cassette for substitution (CmR for UPEC16 Δ*hipH*, GmR for UPEC16 Δ*araD-B*),

and a 500 bp downstream region of the target DNA sequence were fused through fusion PCR. The resulting fusion fragments were then transformed into electrocompetent cells along with a recombineering helper plasmid, pSim6 (*Datta et al., 2006*). After a recovery period of 3–5 hr, the transformed cells were plated on selection plates containing the corresponding antibiotics. the antibiotic-selected recombinant colonies were subsequently validated through PCR analysis and sequencing. Finally, the correct recombinants were incubated at 42 °C to eliminate the temperature-sensitive helper plasmid pSim6.

## Recombinant plasmid construction

Construction of recombinant plasmids was performed using the 2×MultiF Seamless Assembly Mix (ABclonal, RK21020). To tag HipH, *hipH*, together with its native promoter, was fused with Blue Fluorescent Protein gene *bfp* and inserted into the pBAD backbone by in vitro recombination. For Gam-GFP overexpression, the *gam* gene from bacteriophage Mu was fused with Green Fluorescent Protein gene *gfp* and inserted into a p15A ori plasmid by in vitro recombination. The *gam-gfp* was placed under the control of the Tetracycline induction system (Addgene, pFD152, #125546). Similarly, for *recG* overexpression, the *recG* gene was cloned into a p15A ori plasmid, placed under the control of the T5 promoter-Lac-induction system. To express HipH-His, mCherry-His, DosC, and DgcZ (DgcZ-m), these genes were cloned into pBAD, under the control of the Arabinose-induction promoter. Additionally, for the c-di-GMP sensor system purchased from Addgene (#182291), the plasmid origin was substituted with the p15A ori.

## Static biofilm culture

The methodology employed in this study was adapted from the established one (*Merritt et al., 2005*). Bacterial strains of interest were initially inoculated in a 3-to-5-ml culture in tubes and allowed to grow overnight. Subsequently, the stationary cultures were diluted 1:100 in the desired media and cultured until reaching the exponential phase with an OD600 of 0.3. Next, 2 ml of each diluted culture was transferred into a fresh microtiter dish (Biosharp, Cat#BS-20-GJM), which has not been undergo tissue culture treatment. In cases where the subsequent experiment involves imaging, it is necessary to ensure that the microtiter dish has a glass bottom. The plate was covered and incubated at optimal growth temperature for the desired amount of time. While the lids for the microtiter dishes are reusable, it is important to clean them with 70% (v/v) ethanol and allow them to air-dry prior to each experiment.

For cultures involving PdeH or DgcZ induction, it is important to statically incubate cells in microtiter dish for 2 hr at 37 °C first. This incubation period allows cells adherence to the surface. Subsequently, the inducer arabinose was added to the desired concentration to initiate protein induction with desired duration.

## Antibiotic killing assay and persister counting

For cells from static culture, planktonic cells were directly transferred into new tubes and harvested by centrifugation. The remained microliter dishes were gently washed twice by PBS buffer, and adherent cells were harvested by vigorous pipetting. In other cases, the cells were harvested routinely. Subsequently, the harvested cells directly plated on LB plate for colony-forming unit (CFU) counting without antibiotic challenge.

The harvested cells were washed with PBS buffer and subsequently resuspended into fresh LB containing 150 μg/ml ampicillin and incubated for varying durations (0–4.5 hr) at 37 °C with shaking at 220 rpm. The survived cells after ampicillin challenge were harvested by centrifugation and diluted in PBS, followed by plating on LB plate for overnight culture at 37 °C. Persister counting was performed on the next day. Persister ratios were calculated as the ratio of the CFU after antibiotic treatment to the CFU without antibiotics challenge. At least three biological replicates were performed for each experiment. Averages and standard deviations presented are representative of biological replicates.

## Cell growth potential assay and VBNC determination

To evaluate growth potential, bacterial cells were cultured to exponential phase in LB medium to an OD600 of 0.3. For strains harboring an arabinose-induction system for specific protein expression, 0.002% arabinose (w/v) was added to the medium for 3–4 hr at 37 °C with shaking at 220 rpm. The

cells were then plated for colony-forming unit (CFU) counting on LB plates before and after arabinose induction. The cells with growth potential were determined as the ratio of CFUs after arabinose induction to CFUs before arabinose induction. The dead cells ratio was assessed by staining with PI dye. The ratio of VBNC cells was calculated as 100% minus the sum of the ratios of cells with growth potential and dead cells. Averages and standard deviations presented are representative of three biological replicates.

## Microscopy

### Bright-field and fluorescence imaging

Inverted microscope Nikon (eclipse Ti2) and confocal microscope (Leica, Stellaris 5 WLL) were used in this study. The illumination was provided by different lasers, at wavelengths 405 nm for BFP and Hoechst 33258, 488 nm for GFP and SYTO24, 488 nm (Nikon) or 515 nm (Leica) for mVenus$^{NB}$, 561 nm for propidium iodide (PI) and mScarlet-I, respectively. The fluorescence emission signal was imaged to an sCMOS camera (pco.edge 4.2 bi). Appropriate filter sets were selected for each fluorophore according to their spectrum. Image analysis was done by ImageJ software (Fiji). For c-di-GMP sensor analysis, the ratio of mVenus$^{NB}$ to mScarlet-I (R) has a negative correlation with the concentration of c-di-GMP. Therefore, $R^{-1}$ demonstrates a positive correlation with the concentration of c-di-GMP.

### Cells staining

PI, Hoechst 33258 and SYTO24 were added to culture at a final concentration of 1 µg/ml, 10 µg/ml and 10 µM, respectively. The cells were then incubated in the dark at room temperature for 15–30 min. Prior to imaging, the cells were washed twice with PBS.

### Gam-GFP foci imaging

The method employed in this study was adapted from the established one (*Shee et al., 2013*). Cells expressing toxin protein HipH along with DosC or DgcZ under the arabinose-induction system and expressing Gam-GFP under Tetracycline-induction system were cultured to exponential phase in LB medium with supplement of adequate antibiotics for plasmid maintenance. Gam-GFP was induced by 1 µg/ml anhydrotetracycline for 30 min and then HipH and DosC (or DgcZ) were induced by 0.2% (w/v) arabinose for 2.5 hr at 37 °C and 220 rpm. The cells were washed twice using PBS buffer and then visualized under a fluorescence microscope. The Gam-GFP foci were analyzed using ImageJ software from the individual cell intensity histograms. At least four neighboring pixels, each with an individual fluorescence intensity (f) greater than 1.2 times the mean fluorescence intensity of the cell (a), were deemed as foci. Meanwhile, the area occupied by these pixels needed to be less than one-fifth of the total bacterial area.

### Time-lapse imaging

For recording antibiotic killing and bacteria resuscitation processes, cells harboring c-di-GMP sensor in the 24 hr static culture were collected and washed twice with PBS buffer and imaged on a gel-pad containing 3% low melting temperature agarose in PBS (*Pu et al., 2019*; *Pu et al., 2016*). The gel-pad, prepared at the center of the FCS3 chamber as a gel island, was observed under bright field or epifluorescence illumination. Subsequently, the gel-pad was encircled by LB broth containing 150 µg/ml ampicillin for 6 hr at 35 °C. Fresh LB broth was introduced, and the growth medium was refreshed every 3 hr to allow cells to recover sufficiently.

## Fluorescence-activated cell sorting FACS analysis

All samples were analyzed using a Beckman CytoFLEX SRT flow cytometer equipped with a 100 µm nozzle, and normal saline served as the sheath fluid. The c-di-GMP sensor-labeled strain in the 24 hr static biofilm growth phase was washed and resuspended in sterile PBS. Bacterial cells were identified based on FSC (forward scatter) and SSC (side scatter) parameters. Cells were sorted into distinct groups according to their fluorescence intensity (PB450 for BFP, FITC for mVenus$^{NB}$, ECD for mScarlet-I). The obtained results were analyzed using FlowJo V10 software (Treestar, Inc).

## Protein purification

All proteins used in this study were purified through affinity chromatography using a His-tag and a nickel column. Briefly, MG1655 Δ*ara* harboring pBAD::*hipH-GFP*-His (or *GFP*-His) grew to exponential

growth phase in 500 ml LB medium. Induction of HipH-GFP-His (or GFP-His) was carried out by 0.002% arabinose for 2 hr at 37 °C with shaking at 220 rpm. The cells were harvested and resuspended in 25 ml binding buffer (50 mM Tris-HCl, 200 mM NaCl, 10 mM imidazole, 0.5% Triton X-100, pH 8.0) containing 1 mg/ml lysozyme. The samples were incubated at 37 °C for 30 min and then subjected to ultrasonic disruption. The cell lysate was centrifuged at $12,000 \times g$ for 20 min at 4 °C, and the supernatant was collected for protein purification. Ni-TED Sefinose Resin (Sangon Biotech, C610030) was utilized for the purification of His-tagged proteins as per the manual. Finally, HipH-GFP-His (or GFP-His) was stored in elution buffer containing 10% glycerol at –80 °C.

## RNA extraction and real-time quantitative PCR (RT-qPCR) assay

Bacterial strains were grown to mid-exponential growth phase and then induced by 0.002% arabinose for 2 hr. Cells were harvested and washed by PBS twice. Total RNA extraction was carried out using total RNA extraction reagent (Vazyme R403, China), following the manufacture's instruction. Subsequently, 1 µg of total RNA was digested with DNase I and reverse transcribed to cDNA using a reverse transcription kit (Vazyme R223, China). The resulting cDNA was used as template for RT-qPCR (Vazyme Q711, China). The RT-qPCR assay were analyzed using BIO-RAD CFX Connect Real-Time PCR Detection Systems. The 16 S rRNA gene was selected as the internal control. The primers used in RT-qPCR assay were listed in Key resources table. All experiments were performed in biological triplicates to ensure reliability and reproducibility of the results.

## Biotin c-di-GMP pull down assay

The pull-down assay was conducted as previously described (*Chambers and Sauer, 2017*) with some modifications. In the case of using purified proteins, purified HipH-GFP-His or GFP-His protein was added into binding mix containing biotin-c-di-GMP (200 pmol/µl, BIOLOG, B098), 0.2 µl EDTA (200 mM), 2 µl 10×buffer (100 mM Tris-HCl, pH 7.5, 500 mM KCl, 10 mM DTT). The reaction mix was brought to a final volume of 20 µl with the addition of $H_2O$ and then incubated at 25 °C for 20 min. In the case of using cell lysate, MG1655 Δara harboring pBAD::*hipH*-His (or *mCherry*-His) was cultured to exponential phase with an OD600 of 0.3 in 30 ml LB medium. 0.002% arabinose was then added into the culture for another 2 hr at 37 °C with shaking at 220 rpm. The cells were harvested and resuspended in 1.5 ml lysis buffer (TBST containing 1 mg/ml lysozyme, 5 µg/ml DNaseI, 1×protease inhibitor), followed by incubation at 37 °C for 30 min and three cycles of freezing and thawing. The cell lysate was centrifuged at $12,000 \times g$ for 20 min at 4 °C, and the supernatant was collected for the pull-down assay. The specified amount of cell lysate was introduced into a reaction mix containing the indicated amount of biotin-c-di-GMP, 1 µl EDTA (200 mM), and 10 µl 10×buffer. The reaction mix was brought to a final volume of 100 µl with the addition of $H_2O$ and then incubated at 25 °C for 20 min.

Ten µl (per sample) Streptavidin (SA) magnetic beads (Thermo Fisher Scientific, 88816) was washed twice using TBST and resuspended in 150 µl TBST. Washed SA beads were added into reaction buffer and incubated at 25 °C for 1 hr on a rotary table. Subsequently, the SA beads were washed five times using TBST. Biotin-c-di-GMP-conjugated proteins on SA beads were eluted with SDS loading buffer and analyzed by Western blot.

## DNA cleavage assay

For DNA cleavage, 500 ng supercoiled plasmid pBR322 (*Chan et al., 2015*) obtained from the FastPure Plasmid Mini Kit (Vazyme, DC201-01) or relaxed plasmid pBR322 (derived from supercoiled pBR322 digested by Topoisomerase I, Takara, 2240 A), along with 6 µl 5×buffer (170 mM Tris-HCl, pH 7.5, 120 mM KCl, 10 mM DTT, 20 mM MgCl, 25 mM spermidine, 25% Glycerol) and 3 µl BSA (1 mg/ml), were added to the binding mix (as mentioned in Biotin c-di-GMP pull down assay section).

The reaction mix was brought to a final volume of 30 µl with the addition of $H_2O$ and then incubated at 37 °C for varying duration. The reactions were halted by adding SDS to a final concentration of 0.2% and incubated at 37 °C for 20 min. The resulting product DNA was analyzed by 1% (w/v) agarose gel electrophoresis.

## Mouse urinary tract infection

Cells of strains UPEC16-WT and UPEC16 ΔhipH were grew to exponential phase with an OD600 of 0.3, respectively. The cells were harvested and washed with PBS, followed by resuspension in equal

volumes of PBS. Bladder infection was conducted following previously described procedures with slight modifications (*Pang et al., 2022*). Eight-week female BALB/c mice were anesthetized using isoflurane. $2*10^8$ CFU of each strain were transurethrally inoculated by use of 25 G blunt needle. Two rounds of infection were performed with 24 hr interval. At 48 hr post-infection, the mice were sacrificed by cervical dislocation, and the bladders were aseptically removed for following IBC counting assay and persister counting assay. Animal studies were approved by the Animal Care Ethics Committee of Medical Research Institute, Wuhan University (Permit Number: MRI2021-LACA14). We confirm that all experiments conform to the relevant regulatory standards.

## Statistical analysis

Statistical analysis was performed in GraphPad Prism 9 software for Windows. Significance was ascertained by two-tailed Student's t test or Two-way ANOVA. Error bars represent the standard deviations of the mean from at least three independent experiments. A value of $p<0.05$ was considered significant. " * " indicate significant differences (*, $p<0.05$; **, $p<0.01$; ***, $p<0.001$; ****, $p<0.0001$).

## Materials availability

Materials introduced in this study are available from the corresponding author upon request.

## Acknowledgements

We thank Profs. Hongbin Shu (Wuhan University) and Fan Bai (Peking University) for valuable discussions. We also thank the members of our laboratory for helpful discussion. This work is supported by the grants to YP from the National Key R&D Program of China (2021YFC2701602), the Natural Science Foundation of China (31970089), Science Fund for Distinguished Young Scholars of Hubei Province (2022CFA077), Major Project of Guangzhou National Laboratory (GZNL2024A01023), the Fundamental Research Funds for the Central Universities (2042022dx0003). This work is also supported by the grants to YZ for the National Science Fund for Distinguished Young Scholars (82025011) and CG from the Natural Science Foundation of Yunnan Province of China (202001BB050005). We also thank all the staff in the Core Facilities of Medical Research Institute at Wuhan University and the Core Facilities at School of Life Sciences at Peking University for their technical support.

## Additional information

### Funding

| Funder | Grant reference number | Author |
| --- | --- | --- |
| National Key Research and Development Program of China | 2021YFC2701602 | Yingying Pu |
| National Natural Science Foundation of China | 31970089 | Yingying Pu |
| Science Fund for Distinguished Young Scholars of Hubei Province | 2022CFA077 | Yingying Pu |
| Major project of Guangzhou national Laboratory | GZNL2024A01023 | Yingying Pu |
| Fundamental Research Funds for the Central Universities | 2042022dx0003 | Yingying Pu |
| National Science Fund for Distinguished Young Scholars | 82025011 | Yufeng Zhang |

| Funder | Grant reference number | Author |
|---|---|---|
| Natural Science Foundation of Yunnan Province | 202001BB050005 | Chunming Guo |

The funders had no role in study design, data collection and interpretation, or the decision to submit the work for publication.

## Author contributions

Hebin Liao, Conceptualization, Data curation, Formal analysis, Validation, Investigation, Visualization, Methodology, Writing - original draft, Writing - review and editing; Xiaodan Yan, Data curation, Formal analysis, Validation, Investigation, Visualization, Methodology; Chenyi Wang, Formal analysis, Investigation, Visualization, Methodology; Chun Huang, Wei Zhang, Validation, Investigation, Methodology; Leyi Xiao, Jun Jiang, Yongjia Bao, Tao Huang, Hanbo Zhang, Investigation, Methodology; Chunming Guo, Yufeng Zhang, Resources, Funding acquisition; Yingying Pu, Conceptualization, Supervision, Funding acquisition, Writing - original draft, Project administration, Writing - review and editing

## Author ORCIDs

Yufeng Zhang ![ORCID] https://orcid.org/0000-0001-8702-5291
Yingying Pu ![ORCID] https://orcid.org/0000-0002-5735-8199

## Ethics

Animal studies were approved by the Animal Care Ethics Committee of Medical Research Institute, Wuhan University (Permit Number: MRI2021-LACA14). We confirm that all experiments conform to the relevant regulatory standards.

Reviewer #2 (Public Review): https://doi.org/10.7554/eLife.99194.3.sa1
Author response https://doi.org/10.7554/eLife.99194.3.sa2

# Additional files

## Supplementary files

• MDAR checklist

## Data availability

Figure source data files contain the numerical data used to generate the figures.

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

# Appendix 1

## Appendix 1—key resources table

| Reagent type (species) or resource | Designation | Source or reference | Identifiers | Additional information |
|---|---|---|---|---|
| Strain, strain background (*Escherichia coli*) | MG1655 | Yale Genetic Stock Center | CGSC#6300 | |
| Strain, strain background (*Escherichia coli*) | BW25113 | Yale Genetic Stock Center | CGSC#7636 | |
| Strain, strain background (*Escherichia coli*) | UPEC16 | First Affiliated Hospital of Kunming Medical University | | National Genomics Data Center under the accession number "GWHERGI00000000" |
| Strain, strain background (*Escherichia coli*) | BW25113 KEIO library | Dharmacon (GE life sciences) | Cat#OEC4988 | |
| Strain, strain background (*Escherichia coli*) | UPEC16 p15A::c-di-GMP-sensor | This paper | | Figure Legends and Materials and methods section |
| Strain, strain background (*Escherichia coli*) | UPEC16 Δ*ara* pBAD::vector p15A::c-di-GMP-sensor | This paper | | Figure Legends and Materials and methods section |
| Strain, strain background (*Escherichia coli*) | UPEC16 Δ*ara* pBAD::*pdeH* p15A::c-di-GMP-sensor | This paper | | Figure Legends and Materials and methods section |
| Strain, strain background (*Escherichia coli*) | UPEC16 Δ*ara* pBAD::*dgcZ* p15A::c-di-GMP-sensor | This paper | | Figure Legends and Materials and methods section |
| Strain, strain background (*Escherichia coli*) | UPEC16 p15A::c-di-GMP-sensor P-*hipH-bfp* | This paper | | Figure Legends and Materials and methods section |
| Strain, strain background (*Escherichia coli*) | UPEC16 Δ*hipH* | This paper | | Figure Legends and Materials and methods section |
| Strain, strain background (*Escherichia coli*) | MG1655 p15A::c-di-GMP-sensor | This paper | | Figure Legends and Materials and methods section |
| Strain, strain background (*Escherichia coli*) | MG1655 Δ*ara* pBAD::vector p15A::c-di-GMP-sensor | This paper | | Figure Legends and Materials and methods section |
| Strain, strain background (*Escherichia coli*) | MG1655 Δ*ara* pBAD::*dosC* p15A::c-di-GMP-sensor | This paper | | Figure Legends and Materials and methods section |
| Strain, strain background (*Escherichia coli*) | MG1655 Δ*ara* pBAD::*dgcZ* p15A::c-di-GMP-sensor | This paper | | Figure Legends and Materials and methods section |
| Strain, strain background (*Escherichia coli*) | MG1655 Δ*ara* *hipH-bfp* pBAD::vector | This paper | | Figure Legends and Materials and methods section |
| Strain, strain background (*Escherichia coli*) | MG1655 Δ*ara* *hipH-bfp* pBAD::*dgcZ* | This paper | | Figure Legends and Materials and methods section |
| Strain, strain background (*Escherichia coli*) | MG1655 Δ*ara* pBAD::*dgcZ* (or *dgcZ*-m) | This paper | | Figure Legends and Materials and methods section |
| Strain, strain background (*Escherichia coli*) | MG1655 Δ*ara* pBAD::*hipH*-His (or *hipH* mutants) | This paper | | Figure Legends and Materials and methods section |
| Strain, strain background (*Escherichia coli*) | MG1655 Δ*ara* pBAD::*hipH*-mScarlet-I | This paper | | Figure Legends and Materials and methods section |

*Appendix 1 Continued on next page*

*Appendix 1 Continued*

| Reagent type (species) or resource | Designation | Source or reference | Identifiers | Additional information |
|---|---|---|---|---|
| Strain, strain background (*Escherichia coli*) | MG1655 Δ*ara* pBAD::*gfp*-His | This paper | | Figure Legends and Materials and methods section |
| Strain, strain background (*Escherichia coli*) | MG1655 Δ*ara* pBAD::*hipH-gfp*-His | This paper | | Figure Legends and Materials and methods section |
| Strain, strain background (*Escherichia coli*) | MG1655 Δ*ara* pBAD::*hipB*-His | This paper | | Figure Legends and Materials and methods section |
| Strain, strain background (*Escherichia coli*) | MG1655 Δ*ara* pBAD::*mCherry*-His | This paper | | Figure Legends and Materials and methods section |
| Strain, strain background (*Escherichia coli*) | MG1655 Δ*ara* pBAD::*hipH* p15A-T5::vector | This paper | | Figure Legends and Materials and methods section |
| Strain, strain background (*Escherichia coli*) | MG1655 Δ*ara* pBAD::*hipH* p15A-T5::*recG* | This paper | | Figure Legends and Materials and methods section |
| Strain, strain background (*Escherichia coli*) | MG1655 Δ*ara* pBAD::Vector, p15A::*gam*-GFP & ara-*hipH* | This paper | | Figure Legends and Materials and methods section |
| Strain, strain background (*Escherichia coli*) | MG1655 Δ*ara* pBAD::*dosC*, p15A::*gam*-GFP & ara-*hipH* | This paper | | Figure Legends and Materials and methods section |
| Strain, strain background (*Escherichia coli*) | MG1655 Δ*ara* pBAD::*dgcZ*, p15A::*gam*-GFP & ara-*hipH* | This paper | | Figure Legends and Materials and methods section |
| Strain, strain background (*Escherichia coli*) | BW25113 keio mutants pBAD::*dgcZ* | This paper | | Figure Legends and Materials and methods section |
| Strain, strain background (*Escherichia coli*) | BW25113 pBAD::*hipH* | This paper | | Figure Legends and Materials and methods section |
| Strain, strain background (*Escherichia coli*) | BW25113 Δ*recA* pBAD::*hipH* | This paper | | Figure Legends and Materials and methods section |
| Strain, strain background (*Escherichia coli*) | BW25113 Promoter-*rhaB-hipH* pBAD::*dgcZ* | This paper | | Figure Legends and Materials and methods section |
| Antibody | Anti-His-tag-mAb (Mouse Monoclonal) | MBL | Cat#D291-3, RRID: AB_10597733 | WB (1:1000) |
| Antibody | HRP-Goat Anti-Mouse IgG (H+L) (Goat polyclonal) | ABclonal | Cat#AS003; RRID: AB_2769851 | WB (1:10000) |
| Recombinant DNA reagent | p15A::c-di-GMP-sensor | This paper | | p15A ori |
| Recombinant DNA reagent | pBAD::vector | This paper | | Arabinose-induction |
| Recombinant DNA reagent | pBAD::*pdeH* | This paper | | Arabinose-induction |
| Recombinant DNA reagent | pBAD::*dgcZ* (or *dgcZ*-m) | This paper | | Arabinose-induction |
| Recombinant DNA reagent | P-*hipH-bfp* | This paper | | *hipH* native promoter induction |
| Recombinant DNA reagent | pBAD::*dosC* | This paper | | Arabinose-induction |
| Recombinant DNA reagent | pBAD::*hipH*-His (or *hipH* mutants) | This paper | | Arabinose-induction |

*Appendix 1 Continued on next page*

*Appendix 1 Continued*

| Reagent type (species) or resource | Designation | Source or reference | Identifiers | Additional information |
|---|---|---|---|---|
| Recombinant DNA reagent | pBAD::*hipH-mScarlet*-I | This paper | | Arabinose-induction |
| Recombinant DNA reagent | pBAD::*gfp*-His | This paper | | Arabinose-induction |
| Recombinant DNA reagent | pBAD::*hipH-gfp*-His | This paper | | Arabinose-induction |
| Recombinant DNA reagent | pBAD::*mCherry*-His | This paper | | Arabinose-induction |
| Recombinant DNA reagent | p15A::*gam*-GFP & ara-*hipH* | This paper | | Tetracycline induction for Gam-GFP, Arabinose-induction for HipH |
| Recombinant DNA reagent | p15A-T5::vector | This paper | | T5 promoter-Lac-induction |
| Recombinant DNA reagent | p15A-T5::*recG* | This paper | | T5 promoter-Lac-induction |
| Sequence-based reagent | recA-QP1 | This paper | PCR primers | CTGCTGATCTTCATCAACCA |
| Sequence-based reagent | recA-QP2 | This paper | PCR primers | AACAGAGGCGTAGAATTTCAG |
| Sequence-based reagent | sulA-QP1 | This paper | PCR primers | GGGCTTATCAGTGAAGTTGTC |
| Sequence-based reagent | sulA-QP2 | This paper | PCR primers | GGCTAATCTGCATTACTTTCGT |
| Sequence-based reagent | tisB-QP1 | This paper | PCR primers | ATGAACCTGGTGGATATCGC |
| Sequence-based reagent | tisB-QP2 | This paper | PCR primers | TTACTTCAGGTATTTCAGAACAGC |
| Sequence-based reagent | hipH-QP1 | This paper | PCR primers | GATTACTGCACAACACCCTG |
| Sequence-based reagent | hipH-QP2 | This paper | PCR primers | GAAGAACCCACAATTTCTCCTG |
| Sequence-based reagent | 16S-QP1 | This paper | PCR primers | TAGAATTCCAGGTGTAGCGG |
| Sequence-based reagent | 16S-QP2 | This paper | PCR primers | GGGTATCTAATCCTGTTTGCTC |
| Peptide, recombinant protein | HipH-GFP-His | This paper | | Figure Legends and Materials and methods section |
| Peptide, recombinant protein | GFP-His | This paper | | Figure Legends and Materials and methods section |
| Commercial assay or kit | 2×MultiF Seamless Assembly Mix | ABclonal | Cat#RK21020 | |
| Chemical compound, drug | Biotin | Thermo Fisher | Cat#29129 | |
| Chemical compound, drug | Streptavidin Magnetic Beads | Thermo Fisher | Cat#88816 | |
| Chemical compound, drug | 2'-Biotin-16-c-di-GMP | BIOLOG | Cat#B098 | |
| Chemical compound, drug | cycle di-GMP | APExBIO | Cat#B7839 | |
| Chemical compound, drug | Arabinose | Sigma | Cat#V900920 | |
| Chemical compound, drug | IPTG | Sangon Biotech | Cat# A600168 | |

*Appendix 1 Continued on next page*

*Appendix 1 Continued*

| Reagent type (species) or resource | Designation | Source or reference | Identifiers | Additional information |
|---|---|---|---|---|
| Chemical compound, drug | Anhydrotetracycline | MedChemExpress | Cat#HY-118660 | |
| Chemical compound, drug | Ampicillin | Sangon Biotech | Cat#A610028 | |
| Chemical compound, drug | Chloramphenicol | Sangon Biotech | Cat#A600118 | |
| Chemical compound, drug | Gentamicin | Sangon Biotech | Cat#A620217 | |
| Chemical compound, drug | Kanamycin | Sangon Biotech | Cat#A600286 | |
| Software, algorithm | Fiji | GitHub | https://fiji.sc/; RRID:SCR_002285 | |
| Software, algorithm | FlowJo | Treestar, Inc | https://www.flowjo.com/ | |

