## [Editor Report · eLife assessment]

This work describes how the toxin-antitoxin (TA) system, which uses the cyclic di-GMP as an antitoxin, controls both the persistence of antibiotics linked to biofilms and the integrity of the bacterial genome. The authors present **solid** evidence linking cyclic di-GMP and the toxin HipH. The work is **valuable** because it establishes the relationship between bacterial persistence and biofilm resilience, which lays a strong basis for future research on the formation of bacterial biofilms and antibiotic resistance.

---

## [Referee Report · Reviewer #2 (Public Review)]

Summary:

Hebin et al reported a fascinating story about antibiotic persistence in the biofilms. First, they set up a model to identify the increased persisters in the biofilm status. They found that the adhesion of bacteria to the surface leads to increased c-di-GMP levels, which might lead to the formation of persisters. To figure out the molecular mechanism, they screened the *E. coli* Keio Knockout Collection and identified the HipH. Finally, the authors used a lot of data to prove that c-di-GMP not only controls HipH over-expression but also inhibits HipH activity, though the inhibition might be weak.

Strengths:

They used a lot of state-of-the-art technologies, such as single-cell technologies as well as classical genetic and biochemistry approaches to prove the concept, which makes the conclusions very solid. Overall, it is a very interesting and solid story that might attract diverse readers working with c-di-GMP, persisters, and biofilm.

Comments on the revised version:

All my concerns have been addressed.

---

## [Author Response]

The following is the authors’ response to the original reviews.

**eLife assessment**
This preprint explores the involvement of cyclic di-GMP in genome stability and antibiotic persistence regulation in bacterial biofilms. The authors proposed a novel mechanism that, due to bacterial adhesion, increases c-di-GMP levels and influences persister formation through interaction with HipH. While the work may provide useful insights that could attract researchers in biofilm studies and persistence mechanisms, the main findings are inadequately supported and require further validation and refinement in experimental design.

We sincerely thank eLife for the through assessment of our manuscript. We appreciate the constructive criticism and see it as an opportunity to strengthen our research. In response to the reviewers' comments and suggestions, we have made significant improvements to our study. We have refined our experimental design and conducted additional experiments to provide more robust evidence supporting our findings. We believe that with these additional experiments and refinements, our study provides robust evidence for this novel mechanism, contributing significantly to the fields of biofilm research and bacterial persistence.

**Public Reviews:**

**Reviewer #1 (Public Review):**
The authors propose a UPEC TA system in which a metabolite, c-di-GMP, acts as the AT with the toxin HipH. The idea is novel, but several key ideas are missing in regard to the relevant literature, and the experimental design is flawed. Moreover, they are absolutely not studying persister cells as Figure 1b clearly shows they are merely studying dying cells since no plateau in killing (or anything close to a plateau) was reached. So in no way has persistence been linked to c-di-GMP. Moreover, I do not think the authors have shown how the c-di-GMP sensor works. Also, there is no evidence that c-di-GMP is an antitoxin as no binding to HipH has been shown. So at best, this is an indirect effect, not a new toxin/antitoxin system as for all 7 TAs, a direct link to the toxin has been demonstrated for antitoxins.

Thank you for your constructive comments on our manuscript. Your insights have prompted us to revisit our data and experimental design, leading to significant improvements in our study.

(1) Clarification on Persister Cell Detection: We sincerely appreciate your astute observation regarding the interpretation of our killing curve in Figure 1B. Upon careful re-examination, we concur that our initial methodology had limitations in revealing the characteristic biphasic pattern associated with persister cells. To address these limitations, we have implemented two key modifications: shortening the sampling interval and extending the antibiotic treatment duration. These adjustments have resulted in an updated killing curve that now exhibits a more pronounced biphasic pattern and a prominent plateau in the late stage of killing, as illustrated in Response Figure 1. This refined pattern aligns with established characteristics of persister cell behavior in antibiotic tolerance studies, providing a more accurate representation of the persister population dynamics in our experimental system. We believe these methodological enhancements significantly improve the reliability and interpretability of our results, offering a clearer insight into the persister cell phenomenon under investigation.

(2) Validation of c-di-GMP Sensor: We appreciate your point about the c-di-GMP sensor. The c-di-GMP sensor, developed by Howard C. Berg's team, is specifically designed to detect relative intracellular concentrations of c-di-GMP in *Escherichia coli cells*. This capability is crucial for understanding the dynamic regulation of c-di-GMP during bacterial responses to environmental stimuli. We have expanded our explanation of the sensor's detection mechanism in lines 138-146 of the manuscript, detailing how it functions to reflect changes in c-di-GMP levels within the cells accurately. The mechanism operates through a series of signaling events that are initiated when c-di-GMP binds to the sensor, leading to measurable outputs that correlate with intracellular concentrations. Additionally, we have provided a schematic chart in Figure S1B to visually support our description regarding the sensor. This figure demonstrates the sensor's responsiveness and specificity in detecting fluctuations in c-di-GMP levels, effectively linking the signaling molecule to cellular behavior. We hope these additions clarify the role of the c-di-GMP sensor in our research and address your concerns regarding its functionality.

(3) HipH and c-di-GMP Interaction: Our pull-down experiments presented in Figure 5A-E provide robust and compelling evidence for a direct physical interaction between HipH and c-di-GMP, and the effects of their interaction reminiscent of toxin-antitoxin systems. Yet we acknowledge c-di-GMP is not a traditional antitoxin since it is not genetically linked to HipH. We have revised our terminology to "TA-like system" to reflect this difference more accurately.

Weaknesses:(1) L 53: biofilm persisters are no different than any other persisters (there is no credible evidence of any different persister cells) so this reviewer suggests changing 'biofilm persisters' to 'persisters' throughout the text.

Thank you for your thoughtful consideration. Upon careful consideration of the current scientific literature, we agree that there is no substantial evidence supporting a distinct category of persister cells specific to biofilms. We have systematically replaced 'biofilm persisters' with 'persisters' throughout the manuscript.

(2) L 51: persister cells do not mutate and, once resuscitated, mutate like any other growing cell so this sentence should be deleted as it promotes an unnecessary myth about persistence.

We sincerely appreciate your astute observation regarding the inaccuracy in line 51. We have removed the sentence in question from line 51. And we also have thoroughly reviewed the entire manuscript to ensure no similar misconceptions are present elsewhere in the text.

(3) L 69: please include the only metabolic model for persister cell formation and resuscitation here based on single cells (e.g., doi.org/10.1016/j.bbrc.2020.01.102 , https://doi.org/10.1016/j.isci.2019.100792); otherwise, you write as if there are no molecular mechanisms for persistence/resuscitation.

Thank you for your valuable suggestion. We appreciate the opportunity to enhance the scientific context of our manuscript. We have added a brief explanation of how ppGpp mediates ribosome dimerization, leading to persistence, and how its degradation triggers resuscitation [1-3] (lines 68-74). We have described the role of cAMP-CRP in regulating persistence through its effects on metabolism and stress responses [4, 5] (lines 74-78). We also explore potential interactions or synergies between our proposed mechanisms and these established metabolic models [6] (lines 383-409). We believe this revision significantly enhances our manuscript by providing a more accurate representation of the current state of knowledge in the field and demonstrating how our work builds upon and contributes to existing models of bacterial persistence.

(4) The authors should cite in the Intro or Discussion that others have proposed similar novel TAs including a ppGpp metabolic toxin paired with an enzymatic antitoxin SpoT that hydrolyzes the toxin (http://dx.doi.org/10.1016/j.molcel.2013.04.002).

We are grateful for your expertise in pointing out this crucial reference. We sincerely appreciate your suggestion to include the reference to previously proposed novel toxin-antitoxin (TA) systems, particularly the ppGpp-SpoT system [6]. In light of this reference, we have expanded our discussion to include: (1) A brief overview of the ppGpp-SpoT system as a novel TA-like mechanism. (2) Comparisons between the ppGpp-SpoT system and our findings on the HipH-c-di-GMP interaction. (3) Reflections on how these systems challenge and expand traditional definitions of TA systems (lines 383-409). We believe this addition significantly enhances the context and strengthens the rationale for considering the HipH-c-di-GMP interaction as a TA-like system. Thank you for your valuable input in helping us situate our research within the broader landscape of TA system biology.

(5) Figure 1b: there are no results in this paper related to persister cells. Figure 1b simply shows dying cells were enumerated. Hence, the population of stressed cells increased, not 'persister cells' (Figure 1f), in the course of these experiments.

We sincerely appreciate your astute observation regarding the interpretation of our killing curve in Figure 1B. Upon careful re-examination, we concur that our initial methodology had limitations in revealing the characteristic biphasic pattern associated with persister cells. To address these limitations, we have implemented (1) Shortened sampling interval: We have reduced the interval between measurements to one hour. (2) Extended sampling duration: The total duration of sampling has been increased to 6 hours (Response Figure 1). The updated killing curve now exhibits a more pronounced biphasic pattern and a prominent plateau in the late stage of killing: (1) Initial rapid decline: From 0-1hours, we observe a steep decrease in bacterial survival (slope ≈ -3~-1.8); (2) Slower decline phase: From 4.5-6 hours, the rate of decline is markedly reduced (slope ≈ -0.17~-0.06). This pattern aligns more closely with established characteristics of persister cell behavior in antibiotic tolerance studies.

(6) Figure S1: I see no evidence that the authors have shown this c-di-GMP detects different c-di-GMP levels since there appears to be no data related to varying c-di-GMP concentrations with a consistent decrease. Instead, there is a maximum. What are the concentration of c-di-GMP on the X-axis for panels C, D, and E? How were c-di-GMP levels varied such that you know the c-di-GMP concentration?

We appreciate your point about the c-di-GMP sensor. To address this, we have included additional data on the sensor's mechanism and validation. The sensor, developed by Howard C. Berg's team, is designed for detecting intracellular c-di-GMP concentrations in *E. coli* [7].

Sensor Design and Mechanism：The sensor developed for detecting c-di-GMP levels in *Escherichia coli* cells is based on a single fluorescent protein biosensor. The protein includes a Fluorescent Protein Base and a c-di-GMP Binding Domain. The fluorescent protein base is mVenusNB, which is the fastest-folding yellow fluorescent protein (YFP). The c-di-GMP binding domain is the MrkH protein is inserted between Y145 and N146 of mVenusNB. MrkH is a transcription factor with a high affinity for c-di-GMP. When MrkH binds to c-di-GMP, it undergoes a significant conformational change. The amino-terminal domain of MrkH rotates 138° relative to its carboxyl-terminal domain upon c-di-GMP binding.This rotation disrupts the mVenusNB chromophore environment, resulting in reduced fluorescence. The sensor system co-expresses mScarletI, a bright, rapidly folding red fluorescent protein. mScarletI serves as a reference for ratiometric measurements. Such design allows for ratiometric measurement of real-time monitoring of c-di-GMP levels in individual cells and control of variations in protein expression levels between cells. This enables the observation of dynamic changes in c-di-GMP concentration, such as the increase seen after *E. coli* surface attachment.

Functioning and Accuracy: The sensor is designed to detect c-di-GMP in the 100 to 700 nM range, which is the physiological range in *E. coli*. The use of a low copy plasmid for expression ensures detection at low concentrations. The ratio (R) of mVenusNB to mScarletI fluorescence emission is measured for individual cells. The sensor shows at least a twofold dynamic range between low and high c-di-GMP conditions. Cells with low c-di-GMP (expressing phosphodiesterase PdeH) show higher R values compared to cells with high c-di-GMP (expressing constitutively active diguanylate cyclase WspR:D70E). A mutant biosensor (Sensor*) with the R113A mutation in MrkH is used as a control. This mutation eliminates c-di-GMP binding ability, allowing differentiation between specific c-di-GMP effects and other cellular changes.

This biosensor system provides a sophisticated tool for visualizing and quantifying c-di-GMP levels in individual bacterial cells with high sensitivity and temporal resolution. By combining a c-di-GMP-sensitive fluorescent protein with a reference fluorescent protein and utilizing ratiometric analysis, the system can accurately reflect changes in intracellular c-di-GMP levels while controlling for other cellular variables.

We have expanded our explanation of its detection mechanism in lines 138-146 and Figure S1B.

(7) The viable portion of the VBNC population are persister cells so there is no reason to use VBNC as a separate term. Please see the reported errors often made with nucleic acid staining dyes in regard to VBNCs.

We appreciate the opportunity to clarify the distinction between VBNC cells and persister cells in our manuscript. It is essential to recognize that VBNC cells and persister cells represent two fundamentally different states of bacterial dormancy. While both may exhibit viability under certain conditions, persister cells are characterized by their ability to resuscitate and grow when environmental conditions become favorable [8]. In contrast, VBNC cells are in a deep dormant state where they cannot be revived through normal culture conditions [9, 10]. This distinction is critical for accurately representing bacterial survival strategies and population dynamics, which is why we maintain the use of the term VBNC separately from persister cells. We have added related references in lines 259.

Regarding the reported errors associated with nucleic acid staining dyes for identifying VBNC cells, we acknowledge that these methods can exhibit limitations. Specifically, nucleic acid stains may fail to reliably differentiate between metabolically active and inactive cells, leading to inaccuracies in quantifying the true VBNC population [11]. In our study, we have opted to utilize propidium iodide (PI) staining to assess cell viability more accurately, as it effectively distinguishes dead cells from viable cells based on membrane integrity [12]. By employing this methodology, we ensure a more precise estimation of the VBNC proportion without conflating it with persister cell dynamics.

**Reviewer #2 (Public Review):**
Summary:Hebin et al reported a fascinating story about antibiotic persistence in the biofilms. First, they set up a model to identify the increased persisters in the biofilm status. They found that the adhesion of bacteria to the surface leads to increased c-di-GMP levels, which might lead to the formation of persisters. To figure out the molecular mechanism, they screened the *E. coli* Keio Knockout Collection and identified the HipH. Finally, the authors used a lot of data to prove that c-di-GMP not only controls HipH over-expression but also inhibits HipH activity, though the inhibition might be weak.

Thank you for your insightful summary of our research. We greatly appreciate your thoughtful consideration of our work.

Strengths:They used a lot of state-of-the-art technologies, such as single-cell technologies as well as classical genetic and biochemistry approaches to prove the concept, which makes the conclusions very solid. Overall, it is a very interesting and solid story that might attract diverse readers working with c-di-GMP, persisters, and biofilm.Weaknesses:(1) Is HipH the only target identified by screening the *E. coli* Keio Knockout Collection?

We appreciate your inquiry about our screening process and the identification of HipH. We did not screen the entire *E. coli* Keio Knockout Collection. Our approach was more targeted, focusing on mutants relevant to enzyme activity regulation. We selected specific mutants based on their potential involvement in c-di-GMP-mediated regulatory pathways. This focused approach allowed us to efficiently identify candidates likely to be involved in persister formation. Among the screened mutants, HipH emerged as a significant hit. Its identification was particularly noteworthy due to its known role in persister formation and its potential as a regulatory target of c-di-GMP. We acknowledge that our targeted approach may have overlooked other potential candidates. We are considering a more comprehensive screening approach in future studies to identify additional targets.

(2) Since the story is complicated, a diagrammatic picture might be needed to illustrate the whole story. And the title does not accurately summarize the novelty of this study.

Thank you for your valuable feedback. We fully agree with your assessment that a visual representation would greatly enhance the clarity of our complex findings. In response to your suggestion, we have added Response Figure 2 (Fig. 6 in revised manuscript, lines 976-981) to our manuscript. This new figure provides a comprehensive visual summary of the key processes and mechanisms uncovered in our study. This graphic summary provides a clear overview of the interconnected nature of surface adhesion, c-di-GMP signaling, and HipH regulation. It also highlights the complex role of c-di-GMP in persister formation and offers readers a visual aid to better understand the molecular mechanisms underlying our findings.

We sincerely appreciate your thoughtful comment regarding the title and its reflection of the study's novelty. After careful consideration, we believe that our original title adequately captures the essence and significance of our research. We have strived to ensure that it accurately represents the scope and novelty of our work while maintaining clarity and conciseness. Nevertheless, we value your input and thank you for taking the time to provide this feedback, as it encourages us to critically evaluate our presentation.

(3) The ratio of mVenusNB to mScarlet-I (R) negatively correlates with the concentration of c-di-GMP. Therefore, R-1 demonstrates a positive correlation with the concentration of c-di-GMP. Is this method validated with other methods to quantify c-di-GMP, or used in other studies？

We appreciate your point about the c-di-GMP sensor. To address this, we have included additional data on the sensor's mechanism and validation. The sensor, developed by Howard C. Berg's team, is designed for detecting intracellular c-di-GMP concentrations in *E. coli* [7].

Sensor Design and Mechanism：The sensor developed for detecting c-di-GMP levels in *Escherichia coli* cells is based on a single fluorescent protein biosensor. The protein includes a Fluorescent Protein Base and a c-di-GMP Binding Domain. The fluorescent protein base is mVenusNB, which is the fastest-folding yellow fluorescent protein (YFP). The c-di-GMP binding domain is the MrkH protein is inserted between Y145 and N146 of mVenusNB. MrkH is a transcription factor with a high affinity for c-di-GMP. When MrkH binds to c-di-GMP, it undergoes a significant conformational change. The amino-terminal domain of MrkH rotates 138° relative to its carboxyl-terminal domain upon c-di-GMP binding.This rotation disrupts the mVenusNB chromophore environment, resulting in reduced fluorescence. The sensor system co-expresses mScarletI, a bright, rapidly folding red fluorescent protein. mScarletI serves as a reference for ratiometric measurements. Such design allows for ratiometric measurement of real-time monitoring of c-di-GMP levels in individual cells and control of variations in protein expression levels between cells. This enables the observation of dynamic changes in c-di-GMP concentration, such as the increase seen after *E. coli* surface attachment.

Functioning and Accuracy: The sensor is designed to detect c-di-GMP in the 100 to 700 nM range, which is the physiological range in *E. coli*. The use of a low copy plasmid for expression ensures detection at low concentrations. The ratio (R) of mVenusNB to mScarletI fluorescence emission is measured for individual cells. The sensor shows at least a twofold dynamic range between low and high c-di-GMP conditions. Cells with low c-di-GMP (expressing phosphodiesterase PdeH) show higher R values compared to cells with high c-di-GMP (expressing constitutively active diguanylate cyclase WspR:D70). A mutant biosensor (Sensor*) with the R113A mutation in MrkH is used as a control. This mutation eliminates c-di-GMP binding ability, allowing differentiation between specific c-di-GMP effects and other cellular changes.

This biosensor system provides a sophisticated tool for visualizing and quantifying c-di-GMP levels in individual bacterial cells with high sensitivity and temporal resolution. By combining a c-di-GMP-sensitive fluorescent protein with a reference fluorescent protein and utilizing ratiometric analysis, the system can accurately reflect changes in intracellular c-di-GMP levels while controlling for other cellular variables.

We have expanded our explanation of its detection mechanism in lines 138-146 and Figure S1B.

(4) References are missing throughout the manuscript. Please add enough references for every conclusion.

We appreciate your feedback regarding the references in our manuscript. We acknowledge the importance of proper citation to support our conclusions and provide context for our work. In response to your comment, we have conducted a comprehensive review of our manuscript and have significantly enhanced our referencing throughout. We have added appropriate citations to support each key statement and conclusion presented in our study. These additional references provide a robust foundation for our findings and place our work within the broader context of the field. The complete list of all references, including the newly added ones, can be found at the end of this response letter as well as in the revised manuscript.

(5) The novelty of this study should be clearly written and compared with previous references. For example, is it the first study to report the mechanism that the adhesion of bacteria to the surface leads to increased persister formation?

We sincerely appreciate the opportunity to highlight and elaborate the novelty of our research. This study provides novel insights into the relationship between bacterial adhesion to surfaces and the subsequent increase in persister cell formation, which has not been explicitly detailed in previous literature. While existing research has established that biofilms typically harbor higher numbers of persister cells, this investigation not only corroborates that finding but also elucidates the mechanisms through which surface adhesion contributes to this phenomenon.

Past studies have predominantly focused on the general characteristics of persister cells and their role in biofilm resilience and antibiotic tolerance without specifically addressing the mechanistic link between adhesion and persister formation [13, 14]. For instance, previous work has shown that surface attachment leads to changes in metabolic activity and signaling pathways within bacterial cells, which could promote persistence, but it has not definitively established a causal relationship between adhesion and increased persister formation. Our study highlights that the elevation of cyclic di-GMP levels after surface adhesion triggers a cascade of physiological changes that significantly enhance the formation of persister cells. In particular, we report that adhesion-induced signaling pathways promote dormancy and tolerance to antibiotics, marking an important advancement from the previous understanding that treated persister cells might arise from random phenotypic variation during biofilm development. we have expanded our discussion in lines 366-381.

In summary, we believe this study stands as one of the first to clearly delineate the mechanism by which bacterial adhesion leads to increased persister formation, providing a valuable contribution to the current understanding of bacterial persistence and biofilm ecology. Thus, we can assert that our findings are not only novel but also essential for informing future research and therapeutic strategies aimed at managing bacterial infections.

(6) in vitro DNA cleavage assay. Why not use bacterial genomic DNA to test the cleaving of HipH on the bacterial genome?

Thank you for your feedback regarding our experimental approach. The decision of not directly using genomic DNA in our experiments was made after careful consideration. The high molecular weight of genomic DNA, which presents significant challenges in handling and analysis. The difficulty in extracting intact genomic DNA, which could potentially compromise the integrity of our results. The challenges associated with electrophoretic separation of such large DNA molecules, which could limit our ability to accurately interpret the data.

Instead, following established practices in molecular biology research and drawing from similar studies in the field [15-17], we opted to use plasmids as model DNA for our experiments. This approach offers several advantages: Plasmids are smaller and more manageable, making them easier to manipulate in laboratory conditions; They can be more readily extracted in intact form, ensuring the quality of our experimental material; Plasmid DNA is more amenable to electrophoretic separation, allowing for clearer and more precise analysis. Despite their smaller size, plasmids retain many of the key characteristics of genomic DNA that are relevant to our study. We believe this approach provides a robust and reliable model for our research while overcoming the practical limitations associated with genomic DNA. It allows us to investigate the fundamental principles we're interested in, while maintaining experimental feasibility and data integrity. We have added related references in lines 314 and 599.

(7) C-di-GMP -HipH is not a TA, it does not fit in the definition of the TA systems. You can say C-di-gmp is an antitoxin based on your study, but C-di-gmp -HipH is not a TA pair.

We appreciate your insightful feedback regarding the classification of the c-di-GMP-HipH interaction. We acknowledged that while our study suggests c-di-GMP may function as an antitoxin to HipH, the c-di-GMP-HipH pair does not constitute a classical TA system due to the lack of genetic linkage. We have replaced the term "TA system" with "TA-like system" when referring to the c-di-GMP-HipH interaction. This more accurately reflects the nature of their relationship while acknowledging that it differs from traditional TA systems.

**Recommendations for the authors:**

**Reviewer #1 (Recommendations For The Authors):**
(1) Either indent or skip a line to indicate a new paragraph; there is no need to do both.

Thank you for your feedback regarding the formatting of our manuscript. We have revised the formatting throughout the main text by using a single blank line to separate paragraphs, without indentation.

(2) L 77: need to define 'c-di-GMP' without using another abbreviation; please write '3,5-cyclic diguanylic acid', etc.

Thank you for your valuable feedback regarding the proper introduction of abbreviations in our manuscript. We have revised line 86 to introduce the full name of c-di-GMP as "3,5-cyclic diguanylic acid". Following this initial introduction, we consistently use the abbreviation "c-di-GMP" throughout the rest of the manuscript.

**Reviewer #2 (Recommendations For The Authors):**
This is a fascinating story, but the title and the manuscript need careful revision to make it more clear. The novelty and logic are not very easy to follow.(1) Figure 1B, " h" is missing

We sincerely thank you for your attentive review and for pointing out the missing "h" in Figure 1B. We have carefully reviewed and revised the figure legend in Figure 1B. The unit of time has been corrected to include "h" (hours) where appropriate, ensuring consistency and accuracy throughout the figure.

(2) Line 222, the in vivo mice model should be cited with the reference.

Thank you for the reminding. We have cited the following reference related to the mice model (line 231).

Pang Y, et al., (2022) Bladder epithelial cell phosphate transporter inhibition protects mice against uropathogenic *Escherichia coli* infection. Cell reports 39: 110698

References

(1) Wood, T.K. and S. Song, *Forming and waking dormant cells: The ppGpp ribosome dimerization persister model.* Biofilm, 2020. **2**: p. 100018.

(2) Song, S. and T.K. Wood, *ppGpp ribosome dimerization model for bacterial persister formation and resuscitation.* Biochem Biophys Res Commun, 2020. **523**(2): p. 281-286.

(3) Wood, T.K., S. Song, and R. Yamasaki, *Ribosome dependence of persister cell formation and resuscitation.* J Microbiol, 2019. **57**(3): p. 213-219.

(4) Niu, H., J. Gu, and Y. Zhang, *Bacterial persisters: molecular mechanisms and therapeutic development.* Signal Transduct Target Ther, 2024. **9**(1): p. 174.

(5) Mok, W.W., M.A. Orman, and M.P. Brynildsen, *Impacts of global transcriptional regulators on persister metabolism.* Antimicrob Agents Chemother, 2015. **59**(5): p. 2713-9.

(6) Amato, S.M., M.A. Orman, and M.P. Brynildsen, *Metabolic control of persister formation in Escherichia coli.* Mol Cell, 2013. **50**(4): p. 475-87.

(7) Vrabioiu, A.M. and H.C. Berg, *Signaling events that occur when cells of Escherichia coli encounter a glass surface.* Proc Natl Acad Sci U S A, 2022. **119**(6).

(8) Liu, J., et al., *Viable but nonculturable (VBNC) state, an underestimated and controversial microbial survival strategy.* Trends Microbiol, 2023. **31**(10): p. 1013-1023.

(9) Pan, H. and Q. Ren, *Wake Up! Resuscitation of Viable but Nonculturable Bacteria: Mechanism and Potential Application.* Foods, 2022. **12**(1).

(10) Ayrapetyan, M., T. Williams, and J.D. Oliver, *Relationship between the Viable but Nonculturable State and Antibiotic Persister Cells.* J Bacteriol, 2018. **200**(20).

(11) Zhao, S., et al., *Absolute Quantification of Viable but Nonculturable Vibrio cholerae Using Droplet Digital PCR with Oil-Enveloped Bacterial Cells.* Microbiol Spectr, 2022. **10**(4): p. e0070422.

(12) Zhao, S., et al., *Enumeration of Viable Non-Culturable Vibrio cholerae Using Droplet Digital PCR Combined With Propidium Monoazide Treatment.* Front Cell Infect Microbiol, 2021. **11**: p. 753078.

(13) Pan, X., et al., *Recent Advances in Bacterial Persistence Mechanisms.* Int J Mol Sci, 2023. **24**(18).

(14) Patel, H., H. Buchad, and D. Gajjar, *Pseudomonas aeruginosa persister cell formation upon antibiotic exposure in planktonic and biofilm state.* Sci Rep, 2022. **12**(1): p. 16151.

(15) Maki, S., et al., *Partner switching mechanisms in inactivation and rejuvenation of Escherichia coli DNA gyrase by F plasmid proteins LetD (CcdB) and LetA (CcdA).* J Mol Biol, 1996. **256**(3): p. 473-82.

(16) Hockings, S.C. and A. Maxwell, *Identification of four GyrA residues involved in the DNA breakage-reunion reaction of DNA gyrase.* J Mol Biol, 2002. **318**(2): p. 351-9.

(17) Chan, P.F., et al., *Structural basis of DNA gyrase inhibition by antibacterial QPT-1, anticancer drug etoposide and moxifloxacin.* Nat Commun, 2015. **6**: p. 10048.